# Inverse altitude effect disputes the theoretical foundation of stable isotope paleoaltimetry

Zhaowei Jing [1,2,3], Wusheng Yu [1] ✉, Stephen Lewis [4], Lonnie G. Thompson [5], Jie Xu [1,6], Jingyi Zhang [1,6], Baiqing Xu [1], Guangjian Wu [1], Yaoming Ma[1], Yong Wang [1,6] & Rong Guo [1,6]

Stable isotope paleoaltimetry that reconstructs paleoelevation requires stable isotope ($\delta D$ or $\delta^{18}O$) values to follow the altitude effect. Some studies found that the $\delta D$ or $\delta^{18}O$ values of surface isotopic carriers in some regions increase with increasing altitude, which is defined as an "inverse altitude effect" (IAE). The IAE directly contradicts the basic theory of stable isotope paleoaltimetry. However, the causes of the IAE remain unclear. Here, we explore the mechanisms of the IAE from an atmospheric circulation perspective using $\delta D$ in water vapor on a global scale. We find that two processes cause the IAE: (1) the supply of moisture with higher isotopic values from distant source regions, and (2) intense lateral mixing between the lower and mid-troposphere along the moisture transport pathway. Therefore, we caution that the influences of those two processes need careful consideration for different mountain uplift stages before using stable isotope palaeoaltimetry.

Stable isotope paleoaltimetry[1] (following the stable isotopes' altitude effect, i.e., $\delta D$ or $\delta^{18}O$ decreases with increasing altitude) has been widely applied to reconstruct paleoelevation in the Alps[2], the Andes[3], the Rockies[4], and the Tibetan Plateau[5,6]. However, in some situations stable isotope paleoaltimetry has proven unreliable as the results do not match the findings from other independent proxies. For example, paleoaltimetry reconstructions indicate that the Tibetan Plateau (TP) achieved a near-present elevation by the Eocene[5,7] while palaeobotanical[8,9] and paleontological[10,11] studies suggest that the TP did not reach its modern elevation until the Late Miocene. Similar discrepancies exist for data from the Sierra Nevada Mountain range, western United States of America (WUSA)[12]. The occurrence of such discrepancies is because stable isotope paleoaltimetry assumes that the climate conditions have not changed over millions of years[13–15]. Accounting for the effects of paleoclimate changes on stable isotope

paleoaltimetry, Botsyun et al.[15] re-estimated the Eocene elevation of the TP using a numerical model that coupled stable isotopes with atmospheric circulation. Their new approach avoided the flawed assumptions mentioned previously and yielded an elevation less than 3000 m, which agreed with the other independent proxies. Clearly, such flawed assumptions hinder the reliability of stable isotope paleoaltimetry and in part explain the discrepancies of the results between the paleoaltimetry method and the various proxies.

Another potential complication of the paleoaltimetry method relates to the findings that $\delta D$ and $\delta^{18}O$ values of surface isotopic carriers from some regions including meteoric water[16–19], snow[20,21], river water[22–24], ice cores[25], and some biomarkers[26,27] undergo an "inverse altitude effect" (IAE) (i.e., $\delta D$ and $\delta^{18}O$ values increase with increasing altitude) (Fig. 1; Supplementary Table 1). Indeed, the simulated $\delta^{18}O$ values in precipitation across the southern flank of the TP

[1]State Key Laboratory of Tibetan Plateau Earth System, Resources and Environment (TPESRE), Institute of Tibetan Plateau Research, Chinese Academy of Sciences, Beijing 100101, China. [2]Deep-Sea Multidisciplinary Research Center, Pilot National Laboratory of Marine Science and Technology (Qingdao), Qingdao 266237, China. [3]Frontiers Science Center for Deep Ocean Multi-spheres and Earth System, Key Laboratory of Physical Oceanography, Ocean University of China, Qingdao 266100, China. [4]Catchment to Reef Research Group, Centre for Tropical Water and Aquatic Ecosystem Research, James Cook University, Townsville, QLD 4811, Australia. [5]Byrd Polar and Climate Research Center, The Ohio State University, Columbus, OH 43210, USA. [6]University of Chinese Academy of Sciences, Beijing, China. ✉e-mail: yuws@itpcas.ac.cn

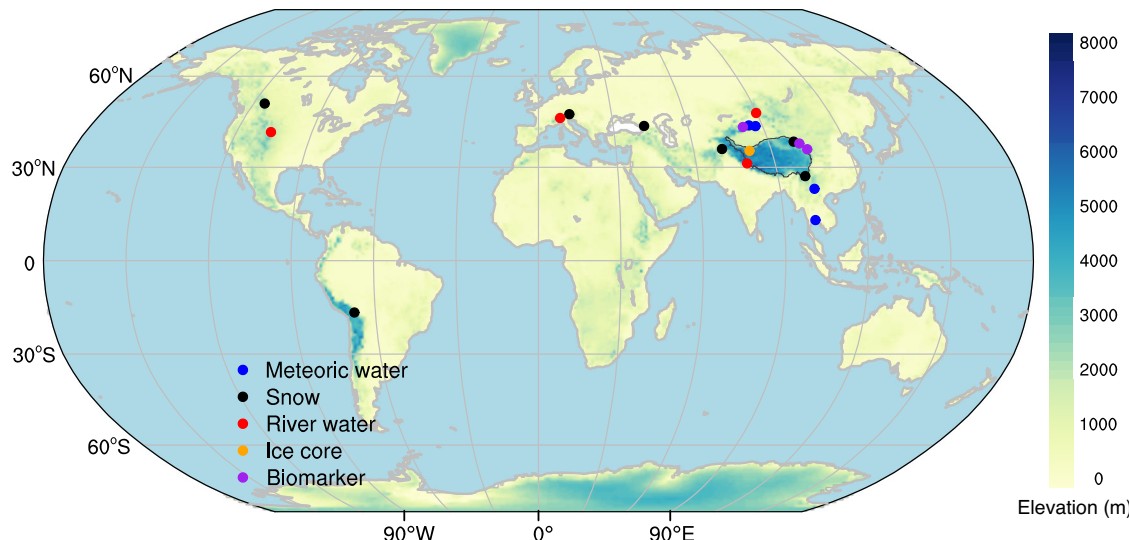

**Fig. 1 | Geographic distribution of the IAE in stable isotopes reported in meteoric water, snow, river water, ice cores, and biomarkers.** Detailed information on these sites is provided in Supplementary Table 1. IAE, inverse altitude effect. This map was generated with The NCAR Command Language (Version 6.6.2) [Software]. (2019). Boulder, Colorado: UCAR/NCAR/CISL/TDD.

during the Eocene also show an IAE[15]. While the flawed assumptions in the paleoaltimetry method lead to increased uncertainties in paleoelevation reconstructions, the IAE directly contradicts the basic theory of stable isotope paleoaltimetry which dictates that δD and δ[18]O values are controlled by the altitude effect (AE)[1]. The IAE has been documented across both observations and simulations, which motivates the need to determine the spatial and temporal variability of the IAE, what causes the IAE to develop and to assess the uncertainty of stable isotope paleoaltimetry reconstructions where it is documented.

Previous studies applied a surface perspective to examine the causes of the IAE at local site scales which have produced diverse findings that are difficult to reconcile. Indeed, post-depositional processes in snow[21,28], sub-cloud evaporation of precipitation[17], local moisture recycling[17], mixing between multiple moisture sources[18], and convective instability[16] have all been proposed to explain the causes of the IAE. A key limitation of these studies is that they mostly focus on a single event and it is unknown whether the IAE holds for the overall climate mean state (seasonal or annual) across these regions. In addition, these studies have focused on measurements from meteoric water or river water sources that contain mixed isotopic signals of water during evaporation, transport, and condensation as well as groundwater pathways and hence they cannot identify the specific hydrological process that controls the IAE[29–31]. These issues highlight the need to better understand the causes of the IAE, especially over larger spatial and temporal scales.

Satellite measurement of stable isotopes in water vapor ($\delta D_v$ or $\delta^{18}O_v$) provides the ability for more targeted examination of the atmospheric hydrological cycle to better quantify the influences on the IAE across larger spatial scales. In particular, the $\delta D_v$ and $\delta^{18}O_v$ above the sub-cloud level are less affected by localized factors that may influence other IAE measurements reported from precipitation and snow records. As such, satellite measurements can be used to independently analyze the influence of large-scale atmospheric circulations on the IAE.

Under different environmental conditions at a specific site, $\delta D_v$ and $\delta^{18}O_v$ generally follow the curves of isotopes in meteoric water[32–35]. The variations of δD are approximately eight times larger than δ[18]O. Therefore, we can apply satellite-derived $\delta D_v$ to identify the key processes responsible for the IAE identified in precipitation δ[18]O and other surface isotopic carriers that are rooted in precipitation. This study reveals the spatial locations and the seasonal

variability of the IAE in $\delta D_v$ retrieved from the Tropospheric Emission Spectrometer (TES) at different atmospheric levels (between 910 and 510 hPa) from 60°S to 60°N. We demonstrate the influence of moisture sources (including the moisture source regions and moisture transport pathways) on the IAE in $\delta D_v$ over the WUSA and the Asian drylands where the IAE is prevalent from an atmospheric circulation perspective. We then determine the connection between the IAE in water vapor and precipitation and discuss implications of the IAE for stable isotope paleoaltimetry. Our results suggest that it is necessary to examine whether the IAE occurs under different topographic scenarios so that the impact of the IAE on stable isotope paleoaltimetry is excluded where otherwise, it will bring great uncertainty to the results of paleoelevation reconstructions.

## Results and discussion
### IAE across the lower and mid-latitudes

Our $\delta D_v$ analysis using TES retrievals from different atmospheric levels across the lower and mid-latitudes (60°S-60°N) found that the IAE mainly occurred across Asia, North Africa, and North America and displayed seasonal patterns with strongest development occurring in the boreal summer (June–July–August: JJA) (Supplementary Figs. 1, 2). Hence, we first focus on the presence of the IAE at different atmospheric levels during summer for those regions.

Between the 750 and 825 hPa levels, the IAE mainly occurred in North Africa and the Asian drylands (from the Red Sea to the northern Tibetan Plateau) during the summer months (Fig. 2, a, e). Between the 681 and 750 hPa levels, the area of the IAE in the Asian drylands increased considerably (Fig. 2f). Although the IAE was almost nonexistent in the WUSA between the 750 and 825 hPa level, it appeared between the 681 and 750 hPa level (Fig. 2b). Between the 618 and 681 hPa levels, the area of the IAE further increased in the WUSA and the northern Tibetan Plateau (NTP) (Fig. 2c, g) while it almost completely disappeared between the 510 and 618 hPa level, with the exception of the NTP region (Fig. 2d, h).

The IAE was weak in the WUSA during the spring (March-April-May: MAM) season (Supplementary Fig. 1b–e) and was not observed during the autumn (September–October–November: SON) and winter (December–January–February: DJF) seasons (Supplementary Fig. 2). However, the IAE persisted in North Africa and the Asian drylands during the spring (Supplementary Fig. 1b–e) and autumn

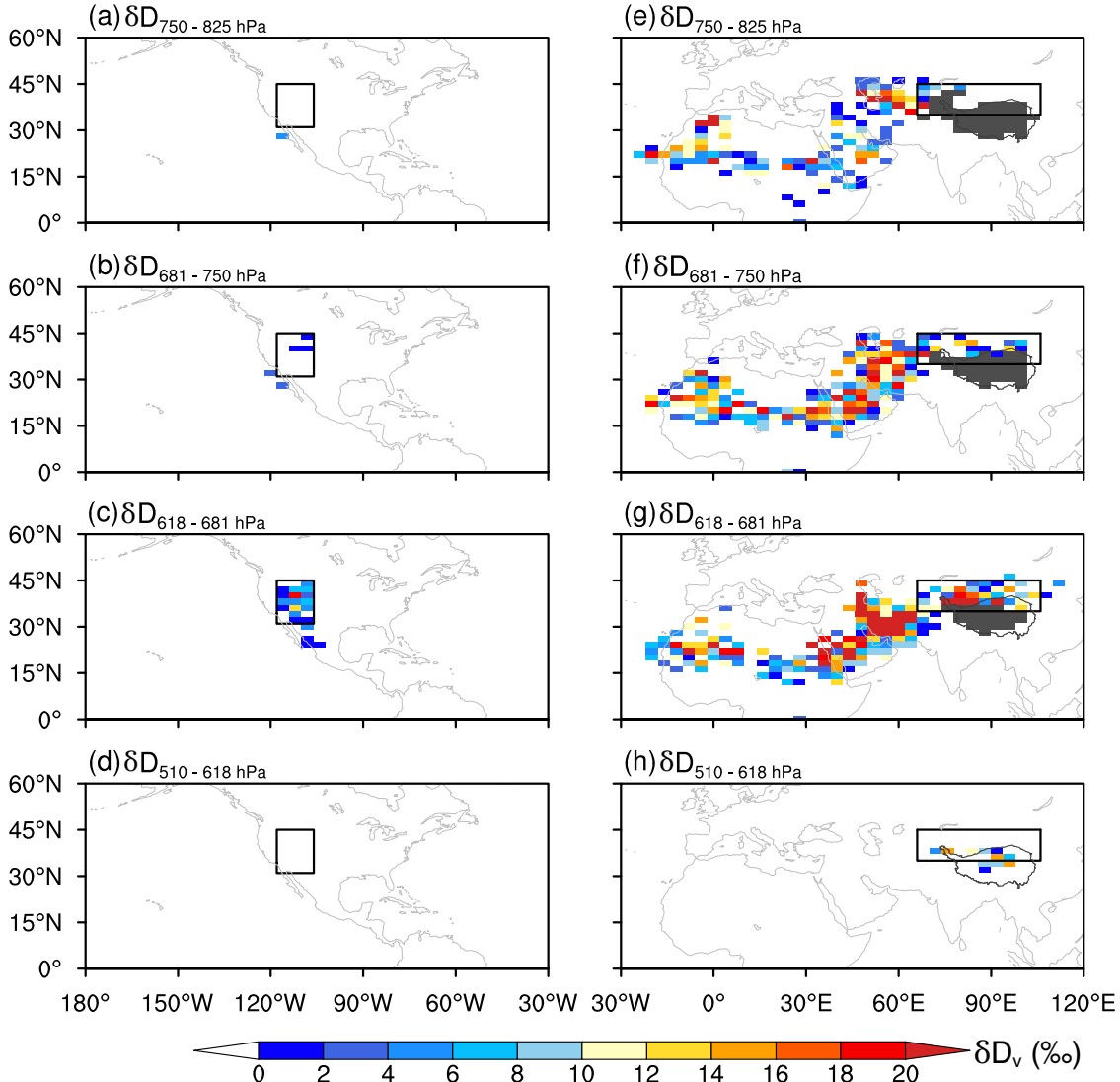

**Fig. 2 | Spatial patterns of the IAE at different atmospheric levels in North America and Afro-Eurasia during the summer months. a-d** IAE over North America and surroundings. **e-h** IAE over Afro-Eurasia and surroundings. The $\delta D_v$ value of the upper level minus the $\delta D_v$ value of the lower level was taken between the two adjacent atmospheric levels. A positive difference indicates that $\delta D_v$ increases with altitude, i.e., the IAE occurs. The black boxes on the left panels show the WUSA region (31° N–45° N, 118° W–106° W) while the black boxes on the right panels indicate the NTP (35° N–45° N, 66° E–106° E). Dark gray lines on the right panels outline the boundary of the Tibetan Plateau. Gray shading on the right panels indicates that no valid $\delta D_v$ data are available for the specific grid. IAE, inverse altitude effect. WUSA, western United States of America. NTP, northern Tibetan Plateau. This map was generated with The NCAR Command Language (Version 6.6.2) [Software]. (2019). Boulder, Colorado: UCAR/NCAR/CISL/TDD.

(Supplementary Fig. 2b–e) seasons. In winter, the IAE diminished greatly over the Asian drylands (Supplementary Fig. 2g–h).

**Causes of the IAE in the western United States of America**

As far as we know neither paleoelevation research nor the study of the IAE have been conducted in North Africa. In contrast, the IAE in surface isotopic carriers such as meteoric water, snow, river water, ice cores, and biomarkers have been widely reported in both the WUSA and the Asian drylands regions. Moreover, these two regions are characterized by high mountains or plateaus such as the Rocky Mountains and the TP (Fig. 1) and provide ideal locations for paleoelevation reconstruction studies[4–6]. Hence, this study primarily focuses on the WUSA and the Asian drylands, although we also provide a brief explanation on the cause of the IAE in North Africa.

The data show that the IAE in the WUSA mainly occurs between the 681 and 618 hPa levels during the summer months but its presence is greatly reduced at the other atmospheric levels (Fig. 2a–d). The IAE in the WUSA over the other seasons becomes greatly diminished

(Supplementary Figs. 1, 2). Previous studies showed that the $\delta D_v$ in the mid-troposphere deviates significantly from the Rayleigh fractionation curve due to mixing of multiple air parcels[36,37]. The summer $\delta-q$ plots for the 825 to 750 hPa level over the WUSA tend to follow the Rayleigh fractionation curve while the $\delta-q$ plots above the 750 hPa level are biased towards the mixing line (Fig. 3a). These results imply that the IAE may be related to the air mixing between multiple moisture sources in the mid-troposphere level.

To further support this inference, we analyzed summer moisture contributions from different atmospheric levels over the moisture source regions to the different atmospheric levels over the target region of the WUSA. Our results show that the moisture transported from the lower troposphere (below 825 hPa) contributes 89% moisture to the target 825 hPa level and 73% to the target 750 hPa level (Supplementary Table 2), which demonstrates that the main moisture transport pathway for the target 825 and 750 hPa levels is characterized by an "upslope" type (Fig. 4a). Under this process, the $\delta D_v$ decreases with increasing altitude due to decreasing air temperature

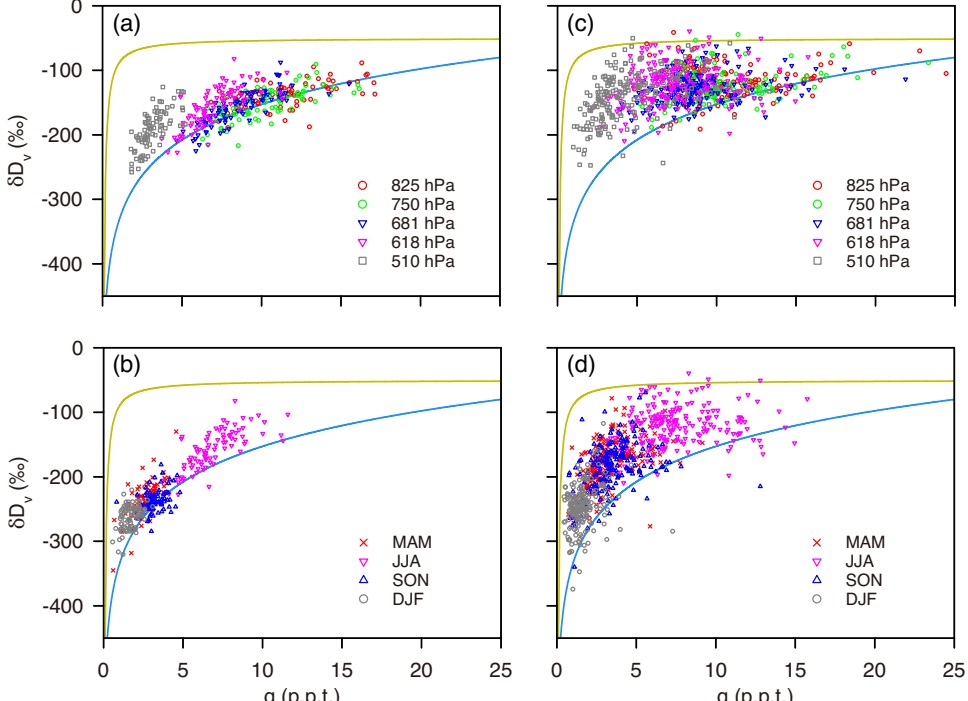

**Fig. 3 | Scatter plots of δD$_v$ versus the water vapor volume mixing ratio q (parts per thousand by volume, p.p.t.). a** Plots of the different atmospheric levels in the WUSA during summer. **b** Plots of the different seasons at the 618 hPa level in the WUSA. **c** Plots of the different atmospheric levels in the NTP during summer. **d** Plots of the different seasons at the 618 hPa level in the NTP. The solid blue curve in each panel represents the Rayleigh fractionation curve calculated for initial conditions of δD$_v$ = −80‰ at $T$ = 25 °C. The solid orange curve in each panel represents the mixing line calculated for two isotopically distinctive air masses (δD$_v$ = −450‰ and δD$_v$ = −80‰) initializing at a water vapor volume mixing ratio of 0.1 p.p.t. WUSA, western United States of America. NTP, northern Tibetan Plateau. MAM, March–April–May (spring); JJA, June–July–August (summer); SON, September-October-November (autumn); DJF, December–January–February (winter).

and follows an "altitude effect". Therefore, the δD$_v$ between the 825 and 750 hPa levels over the WUSA can be well described by the Rayleigh fractionation model. In contrast, the moisture transport pathways are relatively complex from the moisture source regions to the target 681 and 618 hPa levels over the WUSA. While the moisture on route from the lower troposphere contributes 49% and 32% to the target 681 and 618 hPa levels, respectively, the moisture contribution on route from the mid-troposphere becomes the largest (Supplementary Table 2); this finding indicates that the main moisture transport pathway for the target 681 and 618 hPa levels is characterized by an "advection" type (Fig. 4a). We note that the contribution of local vertical mixing on the IAE is very weak for the target 681 and 618 hPa levels (Fig. 4a). Little post-condensation fractionation occurs in the mid-troposphere due to its lower temperature (near or below 0 °C) and as a result, the δD$_v$ at this level is controlled by air mixing and the δD of transported vapor[38]. Previous studies show that mixing processes tend to enrich δD$_v$ in the mid-troposphere more than what would be expected from a Rayleigh distillation process[30,37,39]. Thus, we argue that air mixing is responsible for the IAE between the 681 and 618 hPa levels over the WUSA.

It should be noted that the IAE in the WUSA does not appear at the 510 hPa level in summer despite more intensive lateral mixing at that level (Fig. 3a; Supplementary Table 2). Similarly, the IAE is very weak or does not occur at the 681 and 618 hPa levels in the other seasons (Supplementary Fig. 1c, d; Supplementary Fig. 2c, d, h, i), even though the moisture source predominantly remains from the mid-troposphere (Supplementary Table 3) and the δ–q plots are also biased towards the mixing line (Fig. 3b). These results indicate that lateral mixing is not the sole cause of the IAE in δD$_v$.

The relatively high δD$_v$ values in the mid-troposphere over the WUSA that lead to the development of the IAE indicate that distant moisture with higher isotopic values is carried into this section of the

atmosphere. Hence, we analyzed the seasonal moisture fluxes and δD$_v$ patterns from different atmospheric levels across a large spatial scale (Fig. 5; Supplementary Figs. 4, 5, 6). In the mid-troposphere, the moisture carried by the westerlies is unlikely to cause the IAE in δD$_v$ due to its relatively lower isotopic composition than the WUSA across all four seasons over the period from 2006 to 2009 (Fig. 5h, i; Supplementary Fig. 4h, i; Supplementary Fig. 5h, i; Supplementary Fig. 6h, i). However, another moisture channel derived from the Atlantic anticyclone emerges during summer at the 700 hPa (~681 hPa) and 600 hPa (~618 hPa) levels (Fig. 4a; Fig. 5c, d). The trajectory frequency analyses also confirm the existence of a tropical Atlantic-originated moisture channel that is operational during summer (Supplementary Fig. 7). The δD$_v$ in the mid-troposphere over the tropical Atlantic Ocean in summer is relatively high due to strong convection, which drives this high δD$_v$ moisture from the near-surface into the mid-troposphere over the tropical Atlantic (Fig. 4a; Supplementary Fig. 8b, f). In summer, the oceanic moisture from the 700 and 600 hPa levels over the tropical Atlantic Ocean is laterally transported in a clockwise motion into Mexico through the Caribbean Sea and Gulf of Mexico and then carried northward into the WUSA (Fig. 4a; Fig. 5c, d). Local moisture at the 600 and 700 hPa levels over the WUSA mixes with this moisture from the distant oceanic region (Fig. 5h, i) and results in higher δD$_v$ values within these target atmospheric levels and hence leads to the development of the IAE at the 681 and 618 hPa levels over the WUSA (Figs. 2b, 4a).

The tropical Atlantic-originated moisture channel is relatively weak at the 500 hPa (~510 hPa) level during summer (Fig. 5e); moreover, the moisture from this distant source is characterized by relatively low δD$_v$ values (Fig. 5j). Therefore, the δD$_v$ values at the 510 hPa level compared to the lower sections of the atmosphere explains why the IAE does not occur at this higher level over the WUSA (Fig. 2d).

The anticyclone strength at the 700 and 600 hPa levels over the Atlantic Ocean is weaker (its northward and westward extension is also

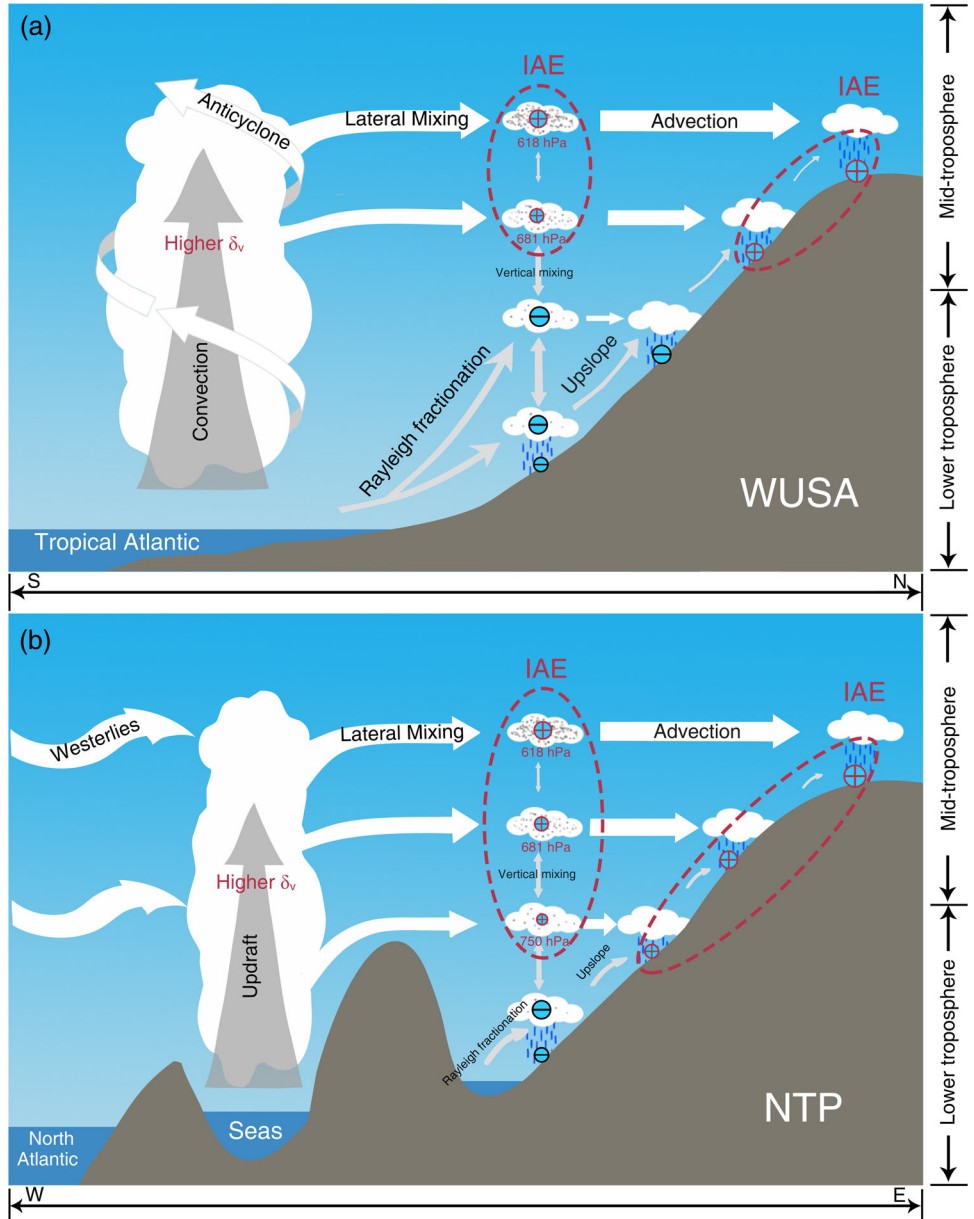

**Fig. 4 | Schematic diagrams of the mechanisms that cause the IAE during summer from an atmospheric circulation perspective. a** WUSA, western United States of America. **b** NTP, northern Tibetan Plateau. Note the plus (minus) signs within the circles indicate the inverse altitude effect (altitude effect), and the sizes of the circles represent the strength of the IAE (AE), i.e., larger circles represent a more pronounced IAE (AE). The white arrows indicate the moisture contributions from the mid-troposphere (above 825 hPa) to the target region. The gray arrows indicate the moisture contributions from the lower troposphere (below 825 hPa) to the target region. The different sizes of the arrows represent the relative moisture contribution percentages. In each panel, red dashed ellipses mark the levels (or altitudes) where the IAE occurs. Seas include the Mediterranean Sea, Red Sea, Persian Gulf, and Caspian Sea. The gray dots in air masses represent the intensity of the lateral mixing, i.e., denser dots, stronger the lateral mixing. IAE, inverse altitude effect. AE, altitude effect. This figure was created with Adobe Photoshop CC 2019.

limited) during the spring and autumn months (Supplementary Figs. 4c, d; 5c, d). Hence, the moisture with higher $\delta D_v$ values from the tropical Atlantic Ocean (Supplementary Fig. 4h, i, Supplementary Fig. 5h, i) is less likely to be transported into the WUSA during spring and autumn, which results in the weakening or disappearance of the IAE at the 681 and 618 hPa levels over the WUSA (Supplementary Fig. 1c, d; 2c, d). In winter, the weak anticyclone strength (Supplementary Fig. 6c, d), coupled with relatively low $\delta D_v$ values over the tropical Atlantic Ocean (Supplementary Fig. 6h, i), explains the lack of IAE in $\delta D_v$ during this season (Supplementary Fig. 2h, i). We therefore conclude that, in addition to intense lateral mixing, the relatively higher isotopic composition of the transported moisture from distant

source regions contributes to the development of the IAE over the WUSA.

### Causes of the IAE in the Asian drylands
Unlike the WUSA, the IAE across the Asian drylands during summer occurs over a wider vertical range from the 750 to 618 hPa levels (Fig. 2e, f, g). Moreover, the IAE in $\delta D_v$ across the Asian drylands occurs not only in summer, but also during spring and autumn (Supplementary Figs. 1, 2).

As the IAE has mainly been reported in the NTP (core region of the Asian drylands) (Fig. 1), we analyzed the $\delta-q$ plots for the different atmospheric levels over the NTP. The $\delta-q$ plots for the different levels

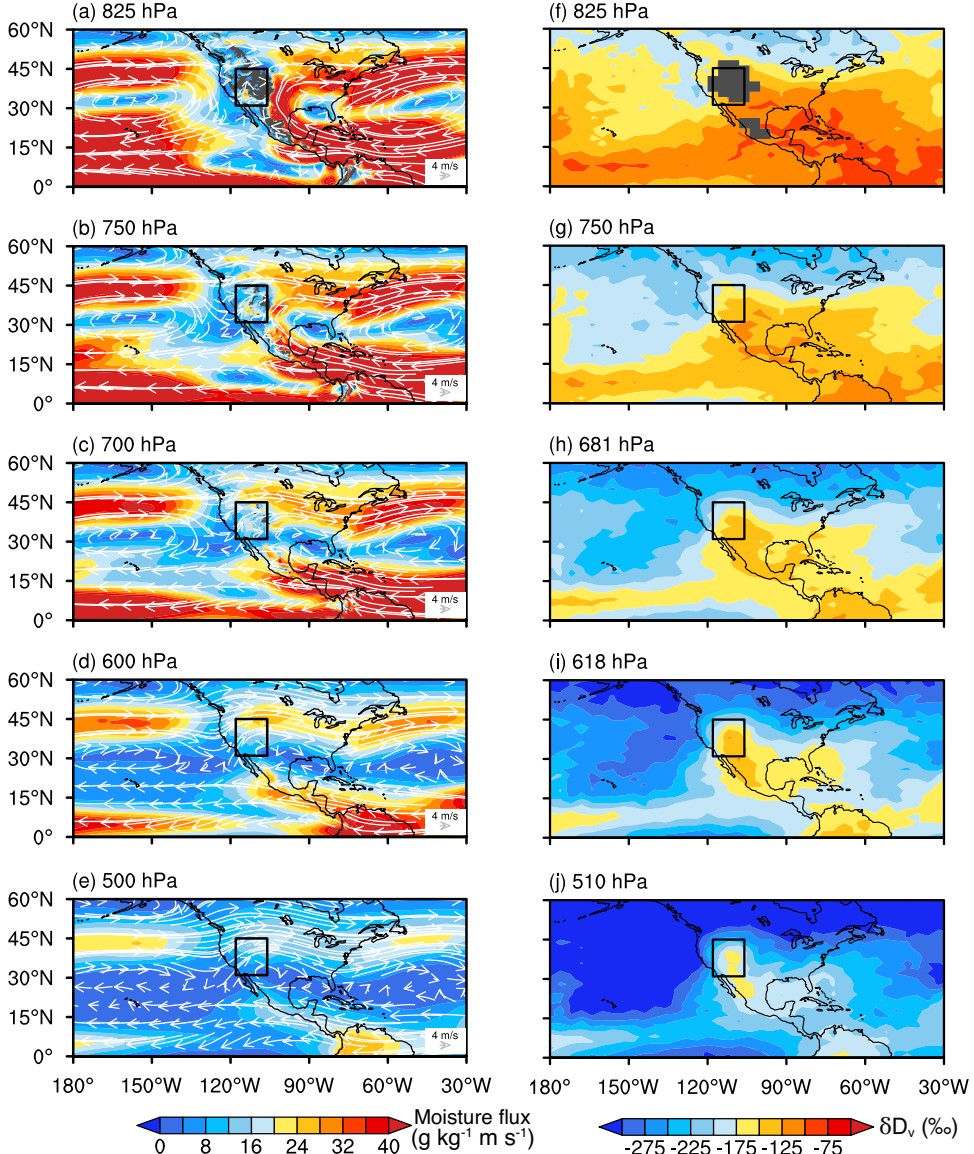

**Fig. 5 | Atmospheric circulation patterns and δD$_v$ during summer (June–July–August: JJA) over North America and surroundings during 2006–2009. a–e** Moisture fluxes (shading) and wind fields (vector) at different atmospheric levels based on ERA5 reanalysis. **f–j** δD$_v$ at the different atmospheric levels based on TES retrievals (right panels). The gray shading in **a**, **b**, **c**, and **f** indicates that no valid δD$_v$ data are available for the specific grid. Black box in each panel represents the WUSA (western United States of America). This map was generated with The NCAR Command Language (Version 6.6.2) [Software]. (2019). Boulder, Colorado: UCAR/NCAR/CISL/TDD.

over the NTP are all biased towards the mixing line (Fig. 3c) which indicate that air mixing is widespread throughout both the lower and mid-troposphere in this region; this finding is also confirmed by moisture transport pathway analysis (Supplementary Table 2). In the NTP, 26% of the moisture at the target 825 hPa level and 41% of the moisture at the target 750 hPa level is derived from the mid-troposphere over the moisture source regions with an "advection" type (above 825 hPa) (Supplementary Table 2; Fig. 4b). This result demonstrates that the moisture at the target 825 and 750 hPa levels over the NTP has already experienced clear lateral mixing within the mid-troposphere along the moisture transport pathway (Supplementary Table 2) and explains why the δ–q plots at these levels are biased towards the mixing line (Fig. 3c). Similar to the WUSA, the moisture for the 681 and 618 hPa levels over the NTP is mainly derived from the 700 hPa level over the moisture source regions (Supplementary Table 2). Therefore, the δ–q plots for the 681 and 618 hPa levels over the NTP are also biased towards the mixing line (Fig. 3c). Based on the δ–q plots and the moisture transport pathway analysis for the NTP, we

argue that the wider vertical range of the IAE across the Asian drylands is the result of widespread lateral mixing.

We find that the IAE in the Asian drylands is greatly diminished at the 510 hPa level during summer (Fig. 2h) and is absent across all atmospheric levels in winter (Supplementary Fig. 2g–j). However, the δ–q plots remain biased towards the mixing line even during the winter months (Fig. 3c, d). Therefore, the occurrence of the IAE is not solely determined by intense lateral mixing. Similar to the findings from the WUSA, the relatively higher isotopic composition of transported moisture from moisture source regions may also be responsible for the development of the IAE in the NTP. To test this hypothesis, we analyzed the seasonal moisture fluxes from different atmospheric levels across a large spatial scale (Supplementary Fig. 9a–e). The data show that the westerlies prevail throughout the year above the 700 hPa level across the region (Supplementary Figs. 9c–e; 10c–e; 11c–e; 12c–e), which is also confirmed by trajectory frequency analyses (Supplementary Fig. 13). During the summer months, several distinctive moisture flux centers develop between the 825 and 600 hPa levels over

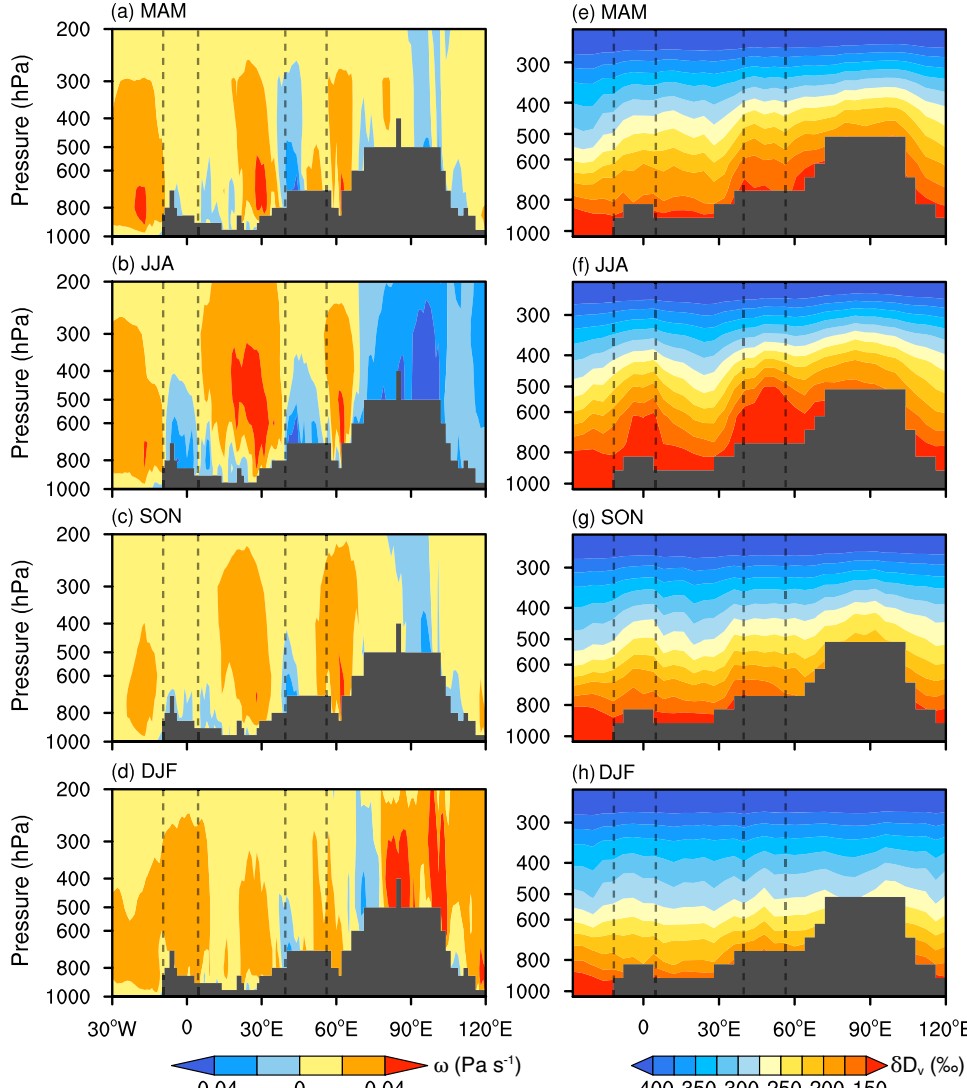

**Fig. 6 | Seasonal vertical profiles of meridional average (15°N–45°N) vertical velocities (ω) and δD_v over the Asian drylands and surroundings during 2006–2009. a–d** Vertical velocities based on ERA5 reanalysis. **e–h** δD_v based on TES retrievals. The dashed lines in each panel represent the longitudinal range of the area where the vertical upward movement is strongest in summer. The gray shading in each panel represents the highest elevation of the surface within a range of longitude, reflecting the complexity of the terrain in each area. MAM, March–April–May (spring); JJA, June–July-August (summer); SON, September–October–November (autumn); DJF, December–January–February (winter).

the Mediterranean Sea, Red Sea, Persian Gulf, and Caspian Sea (Supplementary Fig. 9a–d). Similar moisture flux centers are found during spring and autumn, albeit on a smaller spatial extent (Supplementary Figs. 10a–d; 11a–d). However, the moisture flux centers are not apparent at the 500 hPa level during summer (Supplementary Fig. 9e) and do not occur at any level during winter (Supplementary Fig. 12a–e). Hence, the data indicate that the absence of these moisture flux centers at the 500 hPa level are likely linked to the diminished IAE over the Asian drylands at this level during summer and the absence of the IAE at all levels during winter.

The moisture flux around these centers is very low (Supplementary Fig. 9a–d) which suggests that they are not formed by westerly-dominated moisture transport. In that regard, we contend that these moisture centers are formed by vertical moisture transport from the near surface. To provide support for our contention, we analyzed the vertical profiles of velocity and specific humidity in the Asian drylands and surroundings during 2006–2009 (Fig. 6a–d). Overall, the region is dominated by downward motion throughout the year. However, two distinct upward motion centers develop in the region during the summer months (the range framed by dashed lines in Fig. 6b), which

also coincide with the locations of the moisture flux centers (Supplementary Fig. 9a–d). During summer, the near-surface moisture with higher δD_v is transported upward into the mid-troposphere by updraft. This upward movement results in much higher δD_v values within these two centers compared to the surrounding areas at the same atmospheric level (Figs. 4b; 6f). The moisture from those distant moisture flux centers is transported by the westerlies and horizontally mixed in the mid-troposphere with the moisture over the NTP which results in relatively higher δD_v values at this target atmospheric level over the NTP (Fig. 2e–g; Fig. 4b). If the δD_v values in the levels below the target mid-troposphere are lower, the IAE occurs. Indeed, the moisture with higher δD_v over those centers can be directly transported into the NTP at the 600 hPa (~618 hPa) level compared to the lower 750 and 700 hPa levels as the 600 hPa path is less restricted by the presence of the Pamir-Tianshan Mountains (Supplementary Fig. 9d). As a result, the IAE in δD_v in the NTP is strongest at the target 618 hPa level during summer (Fig. 2g).

Similarly, in the West Asia and North Africa regions, the moisture with relatively higher δD_v values over the distant moisture source regions has a more direct transport pathway at the 750 and 700 hPa

(~681 hPa) levels compared to the lower atmospheric levels which can be blocked by plateaus and mountain ranges (Supplementary Fig. 9b, c). This allows the IAE in $\delta D_v$ to be more obvious at the 750 and 618 hPa levels in those regions (Fig. 2e, f). The same mechanism also explains the occurrence of the IAE during the spring (Fig. 6a, e) and autumn (Fig. 6c, g) months in these regions. In contrast, the weaker upward motion during winter hinders the upward transport of near-surface moisture into the mid-troposphere over those moisture flux centers (Fig. 6d, h) which results in the disappearance of the IAE.

### Connection between the IAE in water vapor and precipitation

As water vapor acts as the "mass source" of precipitation, the stable isotopic composition of water vapor will directly influence the stable isotopic composition of precipitation. Our results indicated that, in the mountainous regions, the patterns of the stable isotopic composition of water vapor at different atmospheric pressure levels govern those of the corresponding precipitation, via advection (Fig. 4). Hence, the IAE in water vapor will be imprinted on precipitation (Fig. 4). Taking the WUSA as an example, at the 618 hPa level, air masses laterally mix with moisture containing higher isotope values along the moisture transport pathway and this signal is preserved through advection processes which results in higher isotope values in precipitation at the target 618 hPa level. In contrast, at the 681 hPa level, the relatively higher isotopic signal in water vapor becomes relatively depleted along the moisture transport pathway due to weak vertical mixing with the lower-troposphere (Supplementary Table 2). This process results in relatively depleted isotope values in precipitation at the 681 hPa level compared to the 618 hPa level, although isotope values in precipitation at the 681 hPa level are still higher than at the lower levels (Fig. 4a). As a consequence, the isotope values in water vapor increase with altitude in the atmosphere from the lower level to the upper level, which produces the IAE in water vapor. Similarly, the isotope values in corresponding precipitation will increase with altitude from the lower topography to the higher topography, and the IAE occurs in corresponding precipitation. Hence, precipitation inherits the IAE in water vapor. Similar processes can be used to explain the IAE in water vapor and precipitation in the NTP (Fig. 4b). The spatial distributions of the IAE reported in precipitation[16–18] and other surface isotopic carriers[20–27] on the global scale (Fig. 1) are mostly consistent with the occurrence of the IAE in water vapor which further demonstrates the close coupling between the stable isotope signals of water vapor and precipitation. It is evident that the IAE in water vapor determines the IAE of precipitation before the influence of localized factors may take part.

### Implications for stable isotope paleoaltimetry

Here we confirm that the IAE also exists in $\delta D_v$, as well as in many surface isotopic carriers. Moreover, we found that in the WUSA and in the NTP, both the moisture supply with relatively higher isotopic values and intense lateral mixing along the moisture transport pathway are indispensable factors for the occurrence of the IAE in $\delta D_v$. It indicates the coupled influence of moisture supply with high isotopic values and intense lateral mixing will disrupt the basic assumptions of stable isotope paleoaltimetry which require isotope values to decrease with increasing altitude. For a mountain range in a specific topographic scenario, where there is no supply of moisture with relatively high isotope values, or no intense lateral mixing between the lower and mid-troposphere along moisture transport pathway, stable isotope paleoaltimetry may still reliably reconstruct paleoelevation. Our study provides a new approach to exclude the adverse effect of the IAE on stable isotope paleoaltimetry. In addition, numerical models that reconstruct paleoelevation suffer from model biases[14,15]. Our results indicate that optimizing the mixing processes between the lower and mid-troposphere in numerical models helps to better constrain the uplift history of mountainous regions.

## Conclusions

Unlike previous studies that attribute the IAE to localized factors[17,21,28], our study takes advantage of the high vertical resolution of $\delta D_v$. These data reveal that a combination of moisture with relatively higher isotopic values from the distant source regions and large-scale lateral mixing between the lower and mid- troposphere generates the IAE in $\delta D_v$. We emphasize that the IAE has already appeared in water vapor before the precipitation event occurs and that the IAE in water vapor will be inherited in precipitation.

The gradual uplift of mountainous regions such as the Tibetan Plateau leads to changes in atmospheric circulation patterns within the broader region, which in turn alters moisture source regions and moisture transport pathways and their inherent patterns in isotopic values. These changes complicate the application of stable isotope paleoaltimetry for such regions. Therefore, it is necessary to verify whether stable isotope paleoaltimetry is valid in different regions under different topographic scenarios, and to exclude the possible influence of the IAE. In addition, climate modeling approaches can suffer from uncertainties in the parameterization of the mixing processes on route between the lower and mid-troposphere during moisture transport. Our results suggest that optimizing the mixing process between the lower and mid-troposphere may significantly improve the model-data agreement on past hydroclimate parameters and better constrain the uplift history of mountain belts like the Tibetan Plateau.

Finally, our study highlights the important roles of air mixing and moisture transport within the mid-troposphere in the isotopic water cycle, which may provide new insights for the research on the causes of the wetter trend in some arid regions[40,41]. We suggest that studying the moisture transport changes from a three-dimensional perspective, rather than from the lower troposphere or total column, may contribute to a more comprehensive understanding of water cycle changes, and a better appreciation of the wetting tendency in some arid regions.

## Methods

### The IAE

Here we focused on the relationships between $\delta D_v$ and the atmospheric pressure levels. Retrievals from the Tropospheric Emission Spectrometer (TES) for the period from 2006 to 2009 were used to analyze the IAE in $\delta D_v$ at the lower and mid-latitudes (60°S–60°N). Firstly, the seasonal averaged $\delta D_v$ for each atmospheric level was calculated separately, and then the $\delta D_v$ of the upper level (higher altitude) minus the $\delta D_v$ of the lower level (lower altitude) was taken between the two adjacent levels. A positive difference in the values between two levels of a grid point indicates that $\delta D_v$ increases with altitude, i.e., the IAE occurs. We note in this study, the definition of the IAE in water vapor is slightly different from the conventional IAE in precipitation or other surface isotopic carriers that traditionally focus on the relationship between isotopes and topography along a mountain range (different locations with increasing altitude). Here we define the IAE in water vapor to describe the relationships between $\delta D_v$ and different atmospheric pressure levels from the same location (same location with increasing altitude), i.e., the $\delta D_v$ increases with increasing altitude in the atmosphere from the lower level to the upper level. However, both terms are used to refer to the variations of isotope with altitude.

### TES retrievals

The TES[29,42], hosted by NASA's Earth Observing System Aura Satellite, is an infrared high-resolution Fourier transform spectrometer with a spectral coverage of 650–3050 cm$^{-1}$ and a spectral resolution of

~0.12 cm$^{-1}$. Tropospheric trace gases, like water vapor and HDO, can be observed globally every 2 days by TES, with a horizontal footprint of 5.3 km × 8.4 km in the nadir viewing mode[42]. The retrieved $\delta D_v$ is most sensitive near the 700 hPa level[42,43]. Moreover, the accuracy of the retrieved $\delta D_v$ is related to temperature, cloud conditions, and water content, so retrievals tend to be more uncertain in the higher latitudes[43,44]. Data with degrees of freedom less than 1.5 and a retrieved quality of 0 were eliminated for data quality control. This filtering procedure is more stringent than previous studies[29,45]. Comparisons with measured data reveal that the overall deviation of retrieved $\delta D_v$ is around 5%[42]. A TES Level 2 lite product (version 7) was used in this study. The effective length of the retrieved $\delta D_v$ was 4 years, from 2006 to 2009, after which instrument degradation problems caused a decrease in sampling quality[42].

### Rayleigh fractionation model

The Rayleigh fractionation and mixing models were used to examine the transport history of moisture. When an air parcel is progressively transported from the moisture source region to the target region, the water vapor within the air parcel preferentially loses the heavier components due to its lower volatility. As a result, the isotopic ratios for the remaining water vapor progressively get further depleted. If we assume that the condensation is immediately removed from the air parcel without exchange with the remaining water vapor, we can define that the air parcel is in Rayleigh condition[46]. Under this condition, the isotopic ratio of the remaining water vapor ($R$) is given by

$$R = R_0 f^{(\alpha-1)} \tag{1}$$

where $R_0$ is the initial isotopic ratio of the water vapor, $f$ is the fraction of the remaining water vapor and $\alpha$ is the fractionation factor between phases. The $\alpha$ is a function of air temperature ($T$, in K)[47] and can be calculated by

$$\ln \alpha = \frac{24.844}{T^2} \times 10^3 - \frac{76.248}{T} + 52.612 \times 10^{-3} \tag{2}$$

### Mixing model

If two air parcels mix with different water vapor volume mixing ratios ($q$), the $q$ of the mixed air parcel ($q_{mix}$) is the weighted average of the $q$ of the two air parcels[30]:

$$q_{mix} = f q_1 + (1-f) q_2 \tag{3}$$

where $f$ is the mixing fraction. According to the law of conservation of mass, the isotopic ratio of the mixed air parcel ($R_{mix}$) is given by

$$R_{mix} = \frac{f[HDO]_1 + (1-f)[HDO]_2}{f[H2O]_1 + (1-f)[H2O]_2} \tag{4}$$

where [HDO] and [H$_2$O] is the isotopic abundance of HDO and H$_2$O in the two air parcels, respectively.

### HYSPLIT back trajectory and moisture source diagnosed methods

The Hybrid Single-Particle Lagrangian Integrated Trajectory model (HYSPLIT) was used to calculate air parcel trajectories[48]. The 240 h back trajectories with 6 h intervals for air parcels at different atmospheric levels (i.e., 825 hPa, 750 hPa, 681 hPa, 618 hPa and 510 hPa) relative to mean-sea-level were determined using ERA-Interim reanalysis[49] with a resolution of 1° × 1°. The HYSPLIT model outputs, including the altitude, latitude, longitude, and specific humidity along the air parcel trajectories, were used for moisture source analysis for each target region. We applied a Lagrangian diagnostic to identify moisture

sources[50]. Note in this study, we use the term moisture source as a relatively broad concept, which includes both the moisture source region and the moisture transport pathway. The changes in the specific humidity ($q_{shum}$) of an air parcel along a trajectory are generally the net result of precipitation and evaporation. An increase of $q_{shum}$ for any 6 h interval, i.e., $\Delta q_{shum} = q_{shum\ (T)} - q_{shum\ (T-6)} > 0$ ($T = 0$ h for start point), indicates that external moisture enters into the air parcel and a moisture uptake event occurs. The location (height, latitude, longitude) at $T$ h is then determined as a moisture source region for the air parcel and $\Delta q_{shum}$ is the initial moisture contribution at the location. When precipitation occurs after the moisture uptake event, moisture contributions from earlier moisture source regions to the target region will decrease proportionally[50]. In this study, we divided the moisture source into eight separate levels to analyze the moisture transport pathway, i.e., 1000 hPa, 900 hPa, 825 hPa, 750 hPa, 700 hPa, 600 hPa, 500 hPa, and 400 hPa, based on the altitude data from the HYSPLIT outputs to better understand the dominant moisture transport pathway to the target region. This study focuses on $\delta D$ in water vapor rather than precipitation, so we present data for all trajectories regardless of whether they produced precipitation within 6 h of the endpoint.

## Data availability

All data used in this study are publicly available. NASA's Jet Propulsion Laboratory provided the TES data (https://tes.jpl.nasa.gov/tes/data). The Copernicus Climate Change Service provided the ERA5 data (https://cds.climate.copernicus.eu/cdsapp#!/home) and ERA-Interim data (https://apps.ecmwf.int/datasets/). Source data for the scatter plots are available from the Supplementary Data file. Source data are provided with this paper.

## Code availability

Data analysis and plotting were performed with NCL (NCAR Command Language, version 6.6.2, https://www.ncl.ucar.edu/).

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

## Acknowledgements

This work was funded by the Second Tibetan Plateau Scientific Expedition and Research Program (STEP) (2019QZKK0201) to W.S.Y., the Basic Science Center for Tibetan Plateau Earth System (BSCTPES, NSFC project no. 41988101-03) to W.S.Y., the National Key R&D Program of China (2017YFA0603303) to W.S.Y., the National Natural Science Foundation of China (42171122 to W.S.Y., 41830964 to Z.W.J.), the Shandong Province's "Taishan" Scientist Project (ts201712017) to Z.W.J., and the Qingdao "Creative and Initiative" frontier Scientist Program (19-3-2-7-zhc) to Z.W.J. We thank John R. Worden from the Jet Propulsion Laboratory for helpful discussion.

## Author contributions

Conceptualization: Z.J., W.Y.; Methodology: Z.J., W.Y.; Visualization: Z.J., J.X.; Supervision: W.Y., S.L., and L.G.T.; Writing—original draft: Z.J., W.Y., S.L., L.G.T., J.Z., B.X., G.W., Y.M., Y.W., and R.G. All authors contributed to interpretation.

## Competing interests

The authors declare no competing interests.
