## [Peer Review File · Nature Communications]

Inverse altitude effect disputes the theoretical foundation of stable isotope paleoaltimetryREVIEWER COMMENTS

Reviewer #1 (Remarks to the Author):

Stable isotopes in carbonates and hydrous silicates are popular proxy records to reconstruct Paleoelevation. Generally, precipitation isotopes in most mountainous regions of the world have significant altitude effects. Therefore, past changes in the isotopic composition of precipitation preserved in pedogenic or authigenic minerals can be used to place numerical constraints on the topographic development of some ancient mountain belts or plateaus. It is very interesting that this study summarized 19 cases of the "inverse elevation effect" (IAE) of precipitation, river water, snow and ice cores. The authors argued that the IAE may resulted in the uncertainty of the paleoaltitude reconstruction based on thermodynamic models, or even shifts the research paradigm of related scientific subjects.

Through the examination of the seasonal IAE in δD_v at different atmospheric levels (between 825 and 510 hPa) from 60°S to 60°N, and analyzing of the δ -q (δ , δD_v ; q, water vapor volume mixing ratio) plots and moisture sources for western United States of America (WUSA) and the Asian drylands, the authors emphasized that the IAE has already appeared in water vapor before the precipitation event occurs. It is believed that "the supply of moisture with higher isotopic values from source regions" and "intense mixing between the lower and mid-troposphere during moisture transport" are the main reasons for IAE phenomenon. At last, they suggested that the optimization of the mixing process between the lower and mid-troposphere may significantly improve the model-data agreement on past hydroclimate parameters and better constrain the uplift history of mountain belts.

The above findings are amazing, it revealed a more universal scientific significance of water related isotopic fields because water vapor is the "source" of land surface and ground water. It may lead to a broader discussion about the IAE phenomenon and its implications not only to paleoaltitude reconstruction, but also to global and local water cycles. The manuscript is well organized and contains all information needed to understand all assumptions, research methods, and the discussion. Conclusions are also clear and well supported by the data obtained from investigations and literature. Some minor revisions are following:

- 1) Since the authors had presented the complex process and mechanism of the existing IAE in water vapor, I suggest to make a graph to show the formation process and influencing factors of "IAE" phenomenon clearly.
- 2) Due to the atmosphere vapor is the "source" of the precipitation, land surface water and ground water, the significance of this findings in this manuscript can contribute to a lot of water isotope-related subjects, not only paleoelevation construction, for example, what are the focuses of future studies on isotopic water cycle?

3)Line 533-547, in table 1, the atmospheric levels of the column is divided as 825 hPa, 750 hPa, 681 hPa, 618 hPa and 510 hPa, while the line is different with it, why? In table 2, please provide the full name of all the abbreviation. In addition, please add a marker to show the atmospheric levels with IAE.

Reviewer #2 (Remarks to the Author):

Thank you for your well written manuscript that deals with the question if an inverse altitude effect is affecting the models used in paleoaltimetry research.

It is an interesting field of research, where still open questions have to be answered. Proxies for assessing paleoaltimetry are quite well understood, but these proxies have to be linked to e.g. stable isotopes assumptions, which are received on the basis of models. The model results of stable isotopes in water vapour or precipitation in e.g. the Eocene cannot be proven by monitoring of stables isotopes in ancient rain. Therefore, it is of importance to have good and reliable models.

The presented results are quite interesting, but I miss the red line in the paper. The introduction starts with paleoaltimetry and the difficulties to model altitudes of mountain regions in former times. Then you argued that there exists the phenomenon of IAE which is a result of 2 factors. ☐ Where is the link to the summary/conclusion of this work?

For 2 regions the IAE is explained in more detail. But it is neither discussed, how these findings will have an impact on paleoaltimetry nor why monitoring studies of stable isotopes often do NOT show an IAE but they rather confirm the common altitude effect. Is the IAE just a phenomenon in water vapour? Have you an impression on how many studies on stable isotopes in meteoric water, snow, glacier, biomarkers and so on show the common altitude affect in respect on the quite small number of studies showing an IAE? Can you specify this? Why don't you link your investigation on IAE to recent findings but on paleoaltimetry-studies?

Additionally, I really miss the discussion of the IAE in North Africa. Your figure 2 shows an occurrence of IAE especially in North Africa, even stronger and with a higher spatial extension than in WUSA. But why don't you analyse these regions? Further, you argue, that IAE can be found in the European Alps, but you do not model any IAE in water vapour for Europe? Can you explain this finding?

The methods section is very short, but in the supplementary material the methods are described quite precise. I suggest to move this section before the results section as it is easier to understand for the reader what is behind the results.

The results section should have another heading, e.g. results and discussion, because it contains not only results. But I really miss a deeper discussion on where does IAE occur and why and what are the consequences on this knowledge for the research community?

The discussion is not a real discussion. It's somehow a summary and too short.

I suggest therefore major revisions before publishing the article.

In the following, there are some more specific acknowledgements:

L. 19: Stable isotopes do not increase. The delta values can increase. Please correct.

L. 24 argument (1): This is not clear for me. When moisture with higher isotopic values is transported to my investigation area, it is supposed to have a depletion in heavy isotopes so the isotope ratio values become lower. This is just the known continental effect. And if moisture with higher isotopic values is transported along mountain ranges, into higher altitudes, there should appear the common altitude effect. So please specify statement (1) in the abstract, what you mean with the source regions and how the IAE effect shall appear.

L. 33: "ref." : This is not a good type of citation. Please specify.

Ll. 42-53: I don't understand this argumentation. Valdes et al. wrote a comment on the article of Botsyun et al. and criticised the published results arguing, that some assumptions in the models were incorrect. I think, you should first discuss, whether the argumentation of Valdes is true or not and why you follow the approach by Botsyun.

L. 101: This is not correct. The pictures are very small and it is difficult to identify the colored fields, but in figure 1 (E) and (I), there are blue and orange dots. Please correct the sentence and enlarge the figures.

L. 256: I miss a citation for the mentioned previous studies,

Reviewer #3 (Remarks to the Author):

Comments to authors

The paper by Jing et al. entitled “Inverse altitude effect disputes the theoretical foundation of stable isotope based paleoaltimetry” examines the cause(s) of the “inverse altitude effect” (IAE) in water vapor in the western United States of America and in Asian drylands where the IAE has already been found in other isotopic carriers. The authors detect two processes that may contribute to this unusual elevation-isotope relationship: (1) the supply of moisture with higher isotopic values from source regions, and (2) intense mixing between the lower and mid-troposphere during moisture transport. The authors suggest that their conclusions are crucial for paleoaltimetry reconstructions and could disturb the “classic” paleoaltimetry approach that takes advantage of isotope decrease with elevation increases.

This is an interesting and important study that definitely deserves to be published in a high-impact journal like Nature Communications. Paleoaltimetry approach based on thermodynamic models (e.g., Rowley et al., 2001) that reconstruct paleoelevation using stable isotopes in carbonates and hydrous silicates are indeed widely used to reconstruct the elevation histories of orogens around the world. However, these models do not account for complex atmospheric processes, such as moisture mixing or change of moisture source, that might change – flatten or even inverse the elevation-isotope relationship. These processes have been hypothesized before to impact water isotopes, however, up to my knowledge, there has been no systematic study of these effects. The study by Jing et al. is therefore of extraordinary importance, and its results are crucial for the further application of the “classical” paleoaltimetry approach. However, I would suggest a moderate revision and have pointed out in my comment below the problems I had reading the text.

My biggest concern is about the definition of the “inverse altitude effect” (IAE). The authors use this term to refer the isotope ratio in the free troposphere. However, the classical paleoaltimetry approach, such as described in Rowley et al., (2001), uses the “isotopic lapse rate”, which is actually an isotope-topography relationship. I think this is a source of a possible speculation in the paper, because the authors first refer to previous works showing the water isotopes decrease with topography increase in surface observations (e.g. snow, ice cores, river water), but then talk about isotope values at different pressure levels in the atmosphere. I think the authors must to make the difference clear and show the connection between the isotope-topography relationship and the relationship in the free troposphere detected in this study. Also, the authors show the existence of IAE at different levels in the free troposphere (Fig. 2. B, C, F, G), but, it seems that IAE does not appear at lower altitude (Fig. 2 A, E). What

is about the sub-cloud level? I think a discussion of the connection between the free-troposphere and the isotopic composition of precipitation must appear in this paper if it is accepted.

My second concern is the correct use of “stable isotopes” in the paper. The introduction part is largely about previous observations and modelling studies on $\delta^{18}\text{O}$ in precipitation ($\delta^{18}\text{O}_p$). However, the study of Jing et al., is about δD in vapour. I would suggest the authors indicate the link between those isotopic species that is not evident for broad audience of the Nature Communications. I suggest that the authors specify exactly which isotopes are involved in the study already the abstract.

Third, I am confused by the term “moisture source”. It is common to trace the “moisture source regions” or “moisture transport pathways” to understand the isotopic signature in vapour (e. g., Sodemann et al., 2009; Dahinden et al., 2021). However, as far as I understand, the authors do not use the term “moisture source” for a geographic region of water uptake, but for different vertical levels of the troposphere (e.g., lines 122-125; Table 1). Isn't this point about moisture uptake at different levels more relevant to prove the increase in vertical mixing in the regions of interest? What about the lateral geographic regions of moisture transport? Maybe it's just me, but I was not clear on what the term “moisture source” actually meant in this study, please explain in more detail in the main text. I find the author's point regarding the change in moisture sources crucial to understanding the paper, as it provides a mechanism to explain the IAE. I would suggest showing the results of the HYPSPPLIT trajectory analysis in the main text?

Finally, the authors attribute the observed IAE to two processes: (1) the supply of moisture with higher isotopic values from source regions, and (2) intense mixing between the lower and mid-troposphere during moisture transport. However, the impact of other processes that might also have a contribution has not been evaluated, only mentioned (lines 65-67). How much these effects, and others (?) contribute to the detected relationships? Please see Risi et al., 2019 (Atmos. Chem. Phys.) for a decomposition method.

Minor points

Line 37. The authors might prefer to refer the latest work on the Alpine paleoaltimetry by Krsnik et al. 2021 (Solid Earth), which builds on the older study by Campany et al. 2012.

Line 57. Some additional references missing from the Introduction: Levin et al., 2009, J. Geophys. Res. Atmos.) and Rohrmann et al. 2014 (EPSL). Both papers found flat isotopic lapse rates.

Line 67. In the study by Levin et al. 2009, the unusual isotopic-altitude relationship was attributed to convective instability in areas of high convective precipitation.

Lines 74-75. Please refer to global studies or studies for different geographic regions, not just for the Tibet

Line 75. Delete “moreover”.

Line 79. Why excluding sub-cloud level?

Line 83. I was confused several times while reading, is it vertical transport or advection? Please specify here and elsewhere in the text.

Lines 125-126. Does this rapid vertical exchange occur everywhere on the continents? Please reference global studies, not a study for China.

Lines 129-130. Replace “simulated” with “described” or similar.

Lines 264-265. Does isotopic composition in the free troposphere play a role in precipitation composition? Can you show this by correlation of so?

Lines 274-275. Better “Uncertainties in the parameterization”?

Lines 276-277. What is about the mixing between the boundary layer and the free troposphere?

Sincerely,

Svetlana Botsyun

Replies to Reviewers

(Reviewers' statements in black, and our response in blue)

Line 224, -the line 224 in the previous manuscript.

L371-L377, -the lines 355-363 in the revision.

REVIEWER COMMENTS

Reviewer #1 (Remarks to the Author):

Stable isotopes in carbonates and hydrous silicates are popular proxy records to reconstruct Paleoelevation. Generally, precipitation isotopes in most mountainous regions of the world have significant altitude effects. Therefore, past changes in the isotopic composition of precipitation preserved in pedogenic or authigenic minerals can be used to place numerical constraints on the topographic development of some ancient mountain belts or plateaus. It is very interesting that this study summarized 19 cases of the "inverse elevation effect" (IAE) of precipitation, river water, snow and ice cores. The authors argued that the IAE may resulted in the uncertainty of the paleoaltitude reconstruction based on thermodynamic models, or even shifts the research paradigm of related scientific subjects.

Through the examination of the seasonal IAE in δD_v at different atmospheric levels (between 825 and 510 hPa) from 60°S to 60°N, and analyzing of the δ - q (δ , δD_v ; q , water vapor volume mixing ratio) plots and moisture sources for western United States of America (WUSA) and the Asian drylands, the authors emphasized that the IAE has already appeared in water vapor before the precipitation event occurs. It is believed that "the supply of moisture with higher isotopic values from source regions" and "intense mixing between the lower and mid-troposphere during moisture transport" are the main reasons for IAE phenomenon. At last, they suggested that the optimization of the mixing process between the lower and mid-troposphere may significantly improve the model-data agreement on past hydroclimate parameters and better constrain the uplift history of mountain belts.

The above findings are amazing, it revealed a more universal scientific significance of water related isotopic fields because water vapor is the "source" of land surface and ground water. It may lead to a broader discussion about the IAE phenomenon and its implications not only to paleoaltitude reconstruction, but also to global and local water cycles. The manuscript is well organized and contains all information needed to understand all assumptions, research methods, and the discussion. Conclusions are also clear and well supported by the data obtained from investigations and literature. Some minor revisions are following:

Response: Thank you very much for providing very positive and encouraging comments. Your comments and suggestions greatly improve the quality of our manuscript. We have tried to address all the concerns raised by you and have carefully revised the manuscript as follows.

1) Since the authors had presented the complex process and mechanism of the existing IAE in water vapor, I suggest to make a graph to show the formation process and influencing factors of "IAE" phenomenon clearly.

Response: Thank you for highlighting this issue. It is a very good idea! Following your suggestion, we added a schematic diagram that presents the complex processes and mechanisms of the IAE over the western United States of America (WUSA) and the northern Tibetan Plateau (NTP). Please see the new Fig. 4 in the revised manuscript.

In addition, we added some lines in a new section “Connection between the IAE in water vapor and precipitation” to further describe the schematic diagram. Please see L306-L333 in the revised manuscript.

Fig. 4. Schematic diagrams of the mechanisms that cause the IAE from an atmospheric circulation perspective. (A) WUSA, western United States of America. (B) NTP, northern Tibetan Plateau. Seas include the Mediterranean Sea, Red Sea, Persian Gulf, and Caspian Sea. The gray dots in air masses represent the intensity of the lateral mixing, i.e., denser dots, stronger the lateral mixing.

2) Due to the atmosphere vapor is the “source” of the precipitation, land surface water and ground water, the significance of this findings in this manuscript can contribute to a lot of water isotope-related subjects, not only paleoelevation construction, for example, what are the focuses of future studies on isotopic water cycle?

Response: Thank you very much for pointing out the extended value for our work, which will likely be of interest to scientists in the study of water vapor in the atmospheric field. For example, the cause(s) for the wetting tendencies in some arid regions remains unclear. Following your suggestion, we added one paragraph in the Conclusion section to address this issue, i.e. “*In addition, our study highlights the important roles of air mixing and moisture transport within the mid-troposphere in the isotopic water cycle, which may provide new insights for the research on the causes of the wetter trend in some arid regions^{39,40}. We suggest that studying the moisture transport changes from a three-dimensional perspective, rather than from the lower troposphere or total column, may contribute to a more comprehensive understanding of water cycle changes, and a better appreciation of the wetting tendency in some arid regions.*” Please see L371-L377 in the revised manuscript.

References:

39. Huang, J. et al. Global semi-arid climate change over last 60 years. *Clim. Dyn.* **46**, 1131–1150 (2016).
40. Ahani, H. et al. Non-parametric trend analysis of the aridity index for three large arid and semi-arid basins in Iran. *Theor. Appl. Climatol.* **112**, 553–564 (2013).

3) Line 533-547, in table 1, the atmospheric levels of the column is divided as 825 hPa, 750 hPa, 681 hPa, 618 hPa and 510 hPa, while the line is different with it, why? In table 2, please provide the full name of all the abbreviation. In addition, please add a marker to show the atmospheric levels with IAE.

Response: We apologize for the confusion. The atmospheric levels of the columns refer to the pressure levels of the TES δD_v over the target regions. The TES δD_v data are characterized by 17 separate levels, but here we mainly focused on the levels of 825 hPa, 750 hPa, 681 hPa, 618 hPa, and 510 hPa. However, the levels of the rows refer to the pressure levels of moisture contribution over the moisture source regions. The moisture contributions were calculated using the ERA data. Hence, the moisture contribution levels followed the levels of the ERA reanalysis, i.e., 1000 hPa, 900 hPa, 825 hPa, 750 hPa, 700 hPa, 600 hPa, 500 hPa, and 400 hPa. Note in this revision we have moved Table 1 and Table 2 to the Supplementary Information as Supplementary Table 2 and Supplementary Table 3, respectively. Following your comments, we added one sentence in Supplementary Table S2 to clarify this issue, i.e., “*Moisture contributions from different levels over the moisture source regions to different levels over the target regions (the WUSA and the NTP) during the summer months. The levels of the columns refer to the pressure levels of the TES δD_v over the target regions, while the levels of the rows refer to the pressure levels of moisture contribution over the moisture source regions that follow the levels of the ERA reanalysis.*” Please see L215-L220 in the revised Supplementary Information.

Following your suggestion, we provide the full name of all the abbreviations in Supplementary Table S3, i.e., “*MAM refers to March-April-May (spring season); JJA refers to June-July-August (summer*

season); *SON* refers to September-October-November (autumn season); *DJF* refers to December-January-February (winter season).” Please see L233-L235 in the revised Supplementary Information.

Following your suggestion, we added some markers to highlight the atmospheric levels that have the IAE. Please see Supplementary Table 2 and Supplementary Table 3 in the revised Supplementary Information.

Reviewer #2 (Remarks to the Author):

Thank you for your well written manuscript that deals with the question if an inverse altitude effect is affecting the models used in paleoaltimetry research.

It is an interesting field of research, where still open questions have to be answered. Proxies for assessing paleoaltimetry are quite well understood, but these proxies have to be linked to e.g. stable isotopes assumptions, which are received on the basis of models. The model results of stable isotopes in water vapour or precipitation in e.g. the Eocene cannot be proven by monitoring of stable isotopes in ancient rain. Therefore, it is of importance to have good and reliable models.

Response: Thank you very much for providing very positive and encouraging comments. Your comments and suggestions greatly improve the quality of our manuscript. We have tried to address all the concerns raised by you and have carefully revised the manuscript as follows.

The presented results are quite interesting, but I miss the red line in the paper. The introduction starts with paleoaltimetry and the difficulties to model altitudes of mountain regions in former times. Then you argued that there exists the phenomenon of IAE which is a result of 2 factors. Where is the link to the summary/conclusion of this work?

Response: We thank you for pointing this issue out. We addressed it as follows.

1) Following your comments, we added two sentences in the Introduction section to link our results with paleoaltimetry, i.e., “*The IAE has been documented across both observations and simulations, which motivates the need to determine the spatial and temporal variability of the IAE, what causes the IAE to develop and to assess the uncertainty of stable isotope paleoaltimetry reconstructions where it is documented.*” and “*We then determine the connection between the IAE in water vapor and precipitation and discuss implications of the IAE for stable isotope paleoaltimetry. Our results suggest that it is necessary to examine whether the IAE occurs under different topographic scenarios so that the impact of the IAE on stable isotope paleoaltimetry is excluded where otherwise, it will bring great uncertainty to the results of paleoelevation reconstructions.*” Note the appearance of the last sentence “Our results suggest that...” follows the style of the journal of Nature Communications. Please see L67-L71 and L102-L107 in the revised manuscript.

2) More importantly, we added a new section “*Implications for stable isotope paleoaltimetry*” to further link our results with paleoaltimetry, i.e., “***Implications for stable isotope paleoaltimetry. Here we confirm that the IAE also exists in δD_v , as well as in many surface isotopic carriers. Moreover, we found that in the WUSA and in the NTP, both the moisture supply with relatively higher isotopic values and intense lateral mixing along the moisture transport pathway are indispensable factors for***

the occurrence of the IAE in δD_v . It indicates the coupled influence of moisture supply with high isotopic values with intense lateral mixing will disrupt the basic assumptions of stable isotope paleoaltimetry which require isotope values to decrease with increasing altitude. For a mountain range in a specific topographic scenario, where there is no supply of moisture with relatively high isotope values, or no intense lateral mixing between the lower and mid-troposphere along moisture transport pathway, stable isotope paleoaltimetry may still reliably reconstruct paleoelevation. Our study provides a new approach to exclude the adverse effect of the IAE on stable isotope paleoaltimetry. In addition, numerical models that reconstruct paleoelevation suffer from model biases^{14,15}. Our results indicate that optimizing the mixing processes between the lower and mid-troposphere in numerical models helps to better constrain the uplift history of mountainous regions.” Please see L334-L349 in the revised manuscript.

References:

14. Valdes, P. J. et al. Comment on “Revised paleoaltimetry data show low Tibetan Plateau elevation during the Eocene”. *Science* **365**, eaax8474 (2019).
15. Botsyun, S. et al. Revised paleoaltimetry data show low Tibetan Plateau elevation during the Eocene. *Science* **363**, eaaq1436 (2019).

For 2 regions the IAE is explained in more detail. But it is neither discussed, how these findings will have an impact on paleoaltimetry nor why monitoring studies of stable isotopes often do NOT show an IAE but they rather confirm the common altitude effect. Is the IAE just a phenomenon in water vapour? Have you an impression on how many studies on stable isotopes in meteoric water, snow, glacier, biomarkers and so on show the common altitude affect in respect on the quite small number of studies showing an IAE? Can you specify this? Why don't you link your investigation on IAE to recent findings but on paleoaltimetry-studies?

Response: Thank you for pointing those issues out. We addressed them as follows.

1) With regard to the impact of these findings on paleoaltimetry: Following your comments, we added a new section “*Implications for stable isotope paleoaltimetry*” to discuss how these findings will have an impact on paleoaltimetry. Please see L334-L349 in the revised manuscript.

2) With regard to “nor discussed why monitoring studies of stable isotopes often do NOT show an IAE but they rather confirm the common altitude effect”: of course it is good news for paleoelevation reconstructions if there is no IAE in the stable isotope data. That is, the altitude effect (AE) is the theoretical foundation of stable isotope paleoaltimetry. Moreover, it is not surprising that many monitoring studies confirm the common AE, which is in accordance with the isotopic fractionation principle. It is clear that the AE is a normal phenomenon. That is why the AE is widely found by a large number of studies.

However, the IAE is an abnormal phenomenon. It seems that the IAE not only contravenes the principle of Rayleigh fractionation, but also directly contradicts the theoretical foundation of stable isotope paleoaltimetry. Hence, we should pay more attention to the IAE, rather than AE. Revealing the cause(s) of the IAE and assessing the uncertainty of stable isotope paleoaltimetry reconstructions driven by the IAE are very important for the paleoelevation reconstructions in mountainous regions.

Obviously, extra discussion about the AE may weaken the topic of this manuscript and the motivation of our study.

3) With regard to whether the IAE is just a phenomenon in water vapour: The IAE is not a phenomenon only in water vapour. The IAE also exists in stable isotopes in meteoric water, snow, river water, ice cores, and biomarkers. Please see Fig 1 which summarizes these occurrences.

4) With regard to “how many studies on stable isotopes in meteoric water, snow, glacier, biomarkers and so on show the common altitude affect in respect on the quite small number of studies showing an IAE”: Yes, we have an impression. The altitude effect (AE) in meteoric water, snow, glacier, biomarkers and so on is more common in previous studies than the IAE. As mentioned above, it is not surprising that many monitoring studies confirm the common AE, which is in accordance with the isotopic fractionation principle. However, some studies confirm the IAE in mountainous regions. In this manuscript, we highlighted 19 cases of the IAE globally (Fig 1). As we know, the IAE will dispute the theoretical foundation of stable isotope paleoaltimetry. Hence, instead of discussing the AE which is much more common, well-studied and established, we pay more attention to the IAE, which has received considerably less attention.

5) With regard to “why don’t you link your investigation on IAE to recent findings but on paleoaltimetry-studies”: One of the important applications of the AE of stable isotopes is to provide modern observational support for the stable isotope paleoaltimetry. Reconstructed paleoelevations using the AE of stable isotopes in carbonates and hydrous silicates have been widely applied in the Alps, the Andes, the Rockies, and the Tibetan Plateau. However, the occurrence of the IAE will dispute the theoretical foundation of stable isotope paleoaltimetry. Hence, some recent studies on the IAE that occurs in meteoric water, snow, glacier, biomarkers and so on, also warrants examination on the implications for stable isotope paleoaltimetry. That is why our investigation links with paleoaltimetry-studies.

In fact, we had intended to not only link our investigation on the IAE to the paleoaltimetry-studies, but also to recent findings. Unfortunately, our investigation on the IAE did not link well to recent findings. Following your comments, we addressed this issue as follows.

a) To link our investigation on the IAE to recent findings, we added some lines in the Introduction section, i.e., “*Previous studies applied a surface perspective to examine the causes of the IAE at local site scales which have produced diverse findings that are difficult to reconcile.*”, “*These issues highlight the need to better understand the causes of the IAE, especially over larger spatial and temporal scales.*” and “*We then determine the connection between the IAE in water vapor and precipitation...*” Please see L72-L73, L83-L84, and L102-L103 in the revised manuscript.

b) To further link our investigation on the IAE to recent findings, we added a new figure, i.e., schematic diagram of the mechanisms of the IAE from an atmospheric circulation perspective. Please see the new Fig. 4 in the revised manuscript.

c) More importantly, we added a new section “*Connection between the IAE in water vapor and precipitation*” to further discuss the link of our investigation on the IAE and recent findings. Please

see L306-L333 in the revised manuscript.

Additionally, I really miss the discussion of the IAE in North Africa. Your figure 2 shows an occurrence of IAE especially in North Africa, even stronger and with a higher spatial extension than in WUSA. But why don't you analyse these regions? Further, you argue, that IAE can be found in the European Alps, but you do not model any IAE in water vapour for Europe? Can you explain this finding?

Response: Thank you for pointing those issues out. We have addressed them as follows.

1) With regard to “no discussion of the IAE in North Africa”: In this study, we used the δD in water vapor on the global scale to discuss the IAE, and found the distributions of the IAE in water vapor are mostly consistent with the IAE in precipitation and other surface isotopic carriers (Fig. 1). However, neither the study of the IAE nor the study of paleoelevation reconstruction has been launched in North Africa, where the average elevation of North Africa is about ~750 m. Hence, we only provided a brief discussion on the IAE in North Africa (please see L296-L305 in the revised manuscript), rather than a detailed discussion. That is why we mainly focused on the regions of WUSA and the Asian drylands, where both the study of the IAE and the paleoelevation reconstruction work have been widely conducted. Following your comments, we added sentences to explain why we did not give a detail discussion of the IAE in North Africa, i.e., “*As far as we know neither paleoelevation research nor the study of the IAE have been conducted in North Africa. In contrast, the IAE in surface isotopic carriers such as meteoric water, snow, river water, ice cores, and biomarkers have been widely reported in the WUSA and in the Asian drylands regions. Moreover, these two regions are characterized by high mountains or plateaus such as the Rocky Mountains and the TP (Fig. 1) and provide ideal locations for paleoelevation reconstruction studies⁴⁻⁶. Hence, this study primarily focuses on the WUSA and the Asian drylands, although we also provide a brief explanation on the cause of the IAE in North Africa.*” Please see L132-L139 in the revised manuscript.

References:

4. Chamberlain, C. P. et al. The Cenozoic climatic and topographic evolution of the western North American Cordillera. *Am. J. Sci.* **312**, 213–262 (2012).
5. Rowley, D. B. & Currie, B. S. Palaeo-altimetry of the late Eocene to Miocene Lunpola basin, central Tibet. *Nature* **439**, 677-681 (2006).

2) With regard to “IAE can be found in the European Alps, but you do not model any IAE in water vapour for Europe”: While an IAE has been reported in the snow and river water of the European Alps, our results from the TES data during 2006–2009 do not show it in the European Alps. Frankly speaking, we don't know the reason for this, although an analysis of a longer time series may provide new insights into this region. Thank you very much for raising an interesting and important issue. We would like to further explore this issue in our future work, if we can obtain longer δD_v time series combined with additional data.

The methods section is very short, but in the supplementary material the methods are described quite precise. I suggest to move this section before the results section as it is easier to understand for the reader what is behind the results.

The results section should have another heading, e.g. results and discussion, because it contains not only results. But I really miss a deeper discussion on where does IAE occur and why and what are the consequences on this knowledge for the research community?

Response: Thank you for pointing those issues out. We addressed them as follows.

1) With regard to the position of the Methods section: Following your suggestion, we moved the Methods section from the Supplementary Information to the main text. Please see L380-L463 in the revised manuscript.

2) With regard to the headings of the Results and Discussion sections: Very sorry for that we made a mistake for the framework of the manuscript, especially the headings of Results and Discussion sections. In fact, the two sub-sections on “Causes of the IAE in the western United States of America” and “Causes of the IAE in the Asian drylands” in the Results section should belong to the Discussion section. Following your suggestion, we moved these two sub-sections into the Discussion section. Please see L141-L305 in the revised manuscript.

In addition, we changed the heading of the Discussion section in previous version to “Conclusion” in the revised manuscript. Please see L350 in the revised manuscript.

3) With regard to “no a deeper discussion on where does IAE occur and why”: The IAE mainly occurs in high mountainous regions. Considering that the IAE in surface isotopic carriers has mainly been reported in the WUSA and the NTP (Fig 1), we mainly discussed the IAE in δD_v in the WUSA and the NTP. Hence, we did not provide a deeper discussion on the occurrence of the IAE in North Africa, the Andes, South Africa, and Australia. Following your comments, we added some sentences to give a criterion why we focused on the WUSA and the NTP. Please see L132-139 in the revised manuscript.

4) With regard to “no a deeper discussion on what are the consequences on this knowledge for the research community”: Following your suggestion, we added a new section “Implications for stable isotope paleoaltimetry” to provide a more thorough discussion on what are the consequences on this knowledge for the research community. Please see L334-L349 in the revised manuscript.

The discussion is not a real discussion. It's somehow a summary and too short.

Response: Thank you very much for pointing this issue out. We are sorry that a mistake was made on the framework of the manuscript, especially the headings of the Discussion and Concussion sections. The “Discussion” section should be “Conclusion” section. Therefore, we changed the heading “Discussion” to “Conclusion” and rephrased this section to summarise the findings of our work. Please see L141 and L350 in the revised manuscript.

Following Reviewer #1’s suggestion and your comments, in the Conclusion section, we added some additional sentences to point out the extended value for our work. Please see L371-L377 in the revised manuscript.

I suggest therefore major revisions before publishing the article.

Response: Thank you very much for your constructive suggestions and comments, which greatly improve our manuscript.

In the following, there are some more specific acknowledgements:

L. 19: Stable isotopes do not increase. The delta values can increase. Please correct.

Response: Following your comments, we changed “stable isotopes increase” to “stable isotope (δD or $\delta^{18}O$) values”. Please see L25-26 in the revised manuscript.

L. 24 argument (1): This is not clear for me. When moisture with higher isotopic values is transported to my investigation area, it is supposed to have a depletion in heavy isotopes so the isotope ratio values become lower. This is just the known continental effect. And if moisture with higher isotopic values is transported along mountain ranges, into higher altitudes, there should appear the common altitude effect. So please specify statement (1) in the abstract, what you mean with the source regions and how the IAE effect shall appear.

Response: Thank you for pointing these issues out. We addressed them as follows.

1) With regard to “a depletion in heavy isotopes so the isotope ratio values become lower”: Yes, what you said is the known continental effect, which can be described by the Rayleigh fractionation. During Rayleigh fractionation process, with the preferential removal of heavy isotopes, the isotope values show the continental effect from ocean to inland and an altitude effect from low altitude to high altitude. Rayleigh fractionation assumes an air parcel is isolated and without advection of additional vapor from outside the system.

However, here we emphasized the important role of the supply of moisture with higher isotopic values from distant source regions and the intense lateral mixing that can occur between the lower and mid-troposphere along the moisture transport pathway which influence the δD_v values. In the mid-troposphere, such mixing processes are common. If an air parcel along the transport pathway mixes with another air parcel with higher isotopic values, the resultant mixed air parcel will have higher isotopic values. This is because that when two air parcels mix, the resulting parcel will be less depleted in heavy isotopes than would be expected from a parcel at the same specific humidity subjected to Rayleigh distillation alone (Galewsky and Hurley 2010). Once the moisture with higher isotopic values is transported to the investigation area, it may increase the isotopic values within the investigation area.

2) With regard to the source regions: In this study, we argued that the source region for the WUSA is the distant tropical Atlantic Ocean and the source regions for the NTP are distant seas (which include the Mediterranean Sea, Red Sea, Persian Gulf, and Caspian Sea). We provided a detailed statement of the source regions in main text. Please see L186-L226 and L263-L295 in the revised manuscript. Due to the word limitation of the Abstract, we could only change “source regions” to “distant source regions” to clarify this point. Please see L33 in the revised manuscript.

3) With regard to “how the IAE effect shall appear”: When the supply of moisture with higher isotopic values from distant source regions exists, combined with intense lateral mixing between the lower and mid-troposphere along moisture transport pathway, the IAE shall appear with a pattern that sees δD_v values increase with increasing altitude in the atmosphere from the lower level to the upper level. In the two subsections of “Causes of the IAE in the western United States of America” and

“Causes of the IAE in the Asian drylands”, we discuss “how the IAE effect shall appear”. Please see L142-L266 and L227-L305 in the revised manuscript. Due to the word limitation of the Abstract, we only changed “(1) the supply of moisture with higher isotopic values from source regions, (2) and intense mixing between the lower and mid-troposphere during moisture transport” to “(1) the supply of moisture with higher isotopic values from distant source regions, and (2) intense lateral mixing between the lower and mid-troposphere along the moisture transport pathway” to clarify this point. Please see L32-34 in the revised manuscript.

In addition, to further clarify how the IAE effect shall appear, we provide a more detailed explanation in the new section “Connection between the IAE in water vapor and precipitation” and added a schematic diagram of the mechanisms that cause the IAE from an atmospheric circulation perspective. Please see L306-L333 in the revised manuscript and the new Fig. 4 in the revised manuscript.

References:

Galewsky, J., & Hurley, J. V. (2010). An advection-condensation model for subtropical water vapor isotopic ratios. *Journal of Geophysical Research: Atmospheres*, 115(D16).

L. 33: “ref.” : This is not a good type of citation. Please specify.

Response: In the revised manuscript, we changed the “Thermodynamic models” to “Stable isotope paleoaltimetry”. Following your comments, we changed “Thermodynamic models (e.g. ref. ¹)...” to “Stable isotope paleoaltimetry¹...” Please see L39 in the revised manuscript.

Ll. 42-53: I don't understand this argumentation. Valdes et al. wrote a comment on the article of Botsyun et al. and criticised the published results arguing, that some assumptions in the models were incorrect. I think, you should first discuss, whether the argumentation of Valdes is true or not and why you follow the approach by Botsyun.

Response: Thank you very much for pointing these issues out. We addressed them as follows.

1) With regard to the assumptions in the models: We are afraid that there are some assumptions in the models including the thermodynamic model (i.e. the one applied for the stable isotope paleoaltimetry) and the numerical model, which may cause confusion. Firstly, we will cover the flawed assumptions of thermodynamic model used by other studies, and not the numerical model used by Botsyun. Valdes also mentioned that the thermodynamic models require several assumptions about the sources of moisture, recycling, and atmospheric circulation, which are difficult to assess from data alone. To avoid the flawed assumptions of the thermodynamic model, Botsyun re-estimated the Eocene elevation of the TP using a numerical model that took into account the effects of paleoclimate changes on paleoaltimetry. On this approach, Valdes stated that “We welcome this approach as a way of refining the reliability and accuracy of thermodynamic model (or stable isotope paleoaltimetry).” To avoid the confusion about these two models, we changed “thermodynamic model” to “stable isotope paleoaltimetry”. Please see L39, L42, L49, L51, L56 and L66 in the in the revised manuscript. In this case, the statement “stable isotope paleoaltimetry” will also be consistent with that of the topic of our study.

2) Secondly, on the criticism of the numerical model: The comments of Valdes, using a HadCM3L numerical model, are somewhat reasonable in some aspects, since no numerical model is absolutely

perfect. Valdes et al. (2019) argue that Botsyun's modeling strategy (Botsyun et al., 2019a) is incomplete, given that Botsyun may run an "atmosphere-only climate model with no feedbacks between the atmosphere, vegetation, and ocean." However, with regard to the technical comments by Valdes et al., 2019 to the paper by Botsyun et al., 2019a, Botsyun et al. (2019b) provided some technical responses, and argued that most concerns of Valdes et al. are indeed not applicable. That is, Botsyun et al. (2019b) replied the comments as "The statement regarding the coupling is not rigorously correct. For our study, we ran the FOAM fully coupled model and the vegetation model LPJ forced by an Eocene paleogeography and greenhouse gas concentrations, specifically to capture the first-order response of SSTs and land surface in a coupled (but lower-resolution and not isotope-enabled) framework and incorporate their feedbacks when forcing LMDZiso with these boundary conditions. We argue that this is a major improvement compared to previous studies that have addressed paleo-elevations using isotope-enabled models."

More importantly, in the study of Botsyun et al. (2019a), the LMDZ numerical model reasonably described the effect of mixing processes on isotopic values, as evidenced by the simulated IAE in precipitation.

In fact, using the HadCM3L numerical model, Valdes also simulated higher isotopic values at relatively high altitudes (Fig 1B in Valdes et al., 2019). Moreover, the outputs of Valdes have some similarities with those of Botsyun, with depletion occurring at low surface elevations between 90° and 110°E (Botsyun et al., 2019b). Hence, it is hard to say which numerical model is more reliable. Considering that they used different numerical models to construct the paleoelevation, it is very difficult for us to discuss whether the argumentation of Valdes is true or not in the Introduction section.

3) With regard to why we follow the approach by Botsyun: we followed the approach by Botsyun for the following reasons.

a) As mentioned in the Introduction section, the results of the paleoelevation reconstructed by Botsyun matched those from other independent evidence, like the paleontological and paleobotanical data.

b) Moreover, some scientists found that δD or $\delta^{18}O$ values of surface isotopic carriers in some regions possess an "inverse altitude effect" (IAE). The model outputs of Botsyun et al (2019) also found an IAE appeared across the southern flank of the TP during the Eocene.

c) More importantly, in our manuscript, we found an important role of mixing processes on the IAE, which confirms the hypothesis of Botsyun.

Hence, we followed the approach by Botsyun.

References:

Botsyun, S. et al. Revised paleoaltimetry data show low Tibetan Plateau elevation during the Eocene. *Science* 363, eaaq1436 (2019a).

Botsyun, S. et al. Response to Comment on "Revised paleoaltimetry data show low Tibetan Plateau elevation during the Eocene". *Science* 365, eaax8990 (2019b).

Valdes, P. J. et al. Comment on "Revised paleoaltimetry data show low Tibetan Plateau elevation during the Eocene". *Science* 365, eaax8474 (2019).

L. 101: This is not correct. The pictures are very small and it is difficult to identify the colored fields, but in figure 1 (E) and (I), there are blue and orange dots. Please correct the sentence and enlarge the figures.

Response: Thank you very much for pointing these issues out. We addressed them as follows.

1) With regard to the very small pictures: Following your suggestion, we divided this picture into two and enlarged them. Please see the new Supplementary Fig. 1 and Supplementary Fig. 2 in the revised Supplementary Information.

2) With regard to the existence of the blue and orange dots and “not correct” expression: Following the Reviewer #3’s suggestion, we also analyzed the IAE under 825 hPa. The results showed that the IAE is weak globally under 825 hPa (new Supplementary Fig. 1A,F; new Supplementary Fig. 2A,F). As for the question you raised, it can be found in the new Supplementary Fig. 1 and Supplementary Fig. 2, that a weak IAE exists at the 681 hPa and 681 hPa levels over the WUSA in spring. In autumn and winter, there is no IAE over the WUSA (Supplementary Fig. 2). Hence, we corrected the sentence to “*The IAE was weak in the WUSA during the spring (March-April-May: MAM) season (Supplementary Fig. 1B-E) and was not observed during the autumn (September-October-November: SON) and winter (December-January-February: DJF) seasons (Supplementary Fig. 2).*” Please see L125-L128 in the revised manuscript.

L. 256: I miss a citation for the mentioned previous studies,

Response: Following your suggestion, we added three citations (as follows) for the mentioned previous studies. Please see L351 in the revised manuscript.

References:

- Kong, Y. & Pang, Z. A positive altitude gradient of isotopes in the precipitation over the Tianshan Mountains: Effects of moisture recycling and sub-cloud evaporation. *J. Hydrol.* **542**, 222–230 (2016).
- Li, Z. et al. The stable isotope evolution in Shiyi glacier system during the ablation period in the north of Tibetan Plateau, China. *Quat. Int.* **380**, 262–271 (2015).
- Moser, H. & Stichler, W. Deuterium and oxygen-18 contents as an index of the properties of snow covers. Snow Mechanics Symposium, Proc. Grindelwald Symp. **114**, pp. 122–135 (1974).

Reviewer #3 (Remarks to the Author):

Comments to authors

The paper by Jing et al. entitled “Inverse altitude effect disputes the theoretical foundation of stable isotope based paleoaltimetry” examines the cause(s) of the “inverse altitude effect” (IAE) in water vapor in the western United States of America and in Asian drylands where the IAE has already been found in other isotopic carriers. The authors detect two processes that may contribute to this unusual elevation-isotope relationship: (1) the supply of moisture with higher isotopic values from source regions, and (2) intense mixing between the lower and mid-troposphere during moisture transport.

The authors suggest that their conclusions are crucial for paleoaltimetry reconstructions and could disturb the “classic” paleoaltimetry approach that takes advantage of isotope decrease with elevation increases.

This is an interesting and important study that definitely deserves to be published in a high-impact journal like Nature Communications. Paleoaltimetry approach based on thermodynamic models (e.g., Rowley et al., 2001) that reconstruct paleoelevation using stable isotopes in carbonates and hydrous silicates are indeed widely used to reconstruct the elevation histories of orogens around the world. However, these models do not account for complex atmospheric processes, such as moisture mixing or change of moisture source, that might change – flatten or even inverse the elevation-isotope relationship. These processes have been hypothesized before to impact water isotopes, however, up to my knowledge, there has been no systematic study of these effects. The study by Jing et al. is therefore of extraordinary importance, and its results are crucial for the further application of the “classical” paleoaltimetry approach. However, I would suggest a moderate revision and have pointed out in my comment below the problems I had reading the text.

Response: Thank you very much for providing very positive and encouraging comments. Your comments and suggestions greatly improve the quality of our manuscript. We have tried to address all the concerns raised by you and have carefully revised the manuscript as follows. In addition, following two other Reviewers’ suggestions and comments, in the Conclusion section, we added some lines to point out the extended value for our work. Please see L371-L377 in the revised manuscript.

My biggest concern is about the definition of the “inverse altitude effect” (IAE). The authors use this term to refer the isotope ratio in the free troposphere. However, the classical paleoaltimetry approach, such as described in Rowley et al., (2001), uses the “isotopic lapse rate”, which is actually an isotope-topography relationship. I think this is a source of a possible speculation in the paper, because the authors first refer to previous works showing the water isotopes decrease with topography increase in surface observations (e.g. snow, ice cores, river water), but then talk about isotope values at different pressure levels in the atmosphere. I think the authors must to make the difference clear and show the connection between the isotope-topography relationship and the relationship in the free troposphere detected in this study. Also, the authors show the existence of IAE at different levels in the free troposphere (Fig. 2. B, C, F, G), but, it seems that IAE does not appear at lower altitude (Fig. 2 A, E). What is about the sub-cloud level? I think a discussion of the connection between the free-troposphere and the isotopic composition of precipitation must appear in this paper if it is accepted.

Response: Thank you for pointing those issues out. We addressed them as follows.

1) With regard to making the difference clear in the definitions of the “inverse altitude effect” (IAE) and the “isotopic lapse rate”: The altitude effect is often expressed as an “isotopic lapse rate” and given as a permil change in $\delta^{18}\text{O}$ or δD of precipitation per 100 m of elevation change. Clearly, the “isotopic lapse rate” is the quantitative expression of the “normal” altitude effect. In our manuscript, however, the IAE is used to describe the “abnormal” variations of isotopes in water vapor at different pressure levels. That is, we can discuss the “isotopic lapse rate” only if there is a “normal” altitude effect. However, when an “abnormal” IAE occurs, the “isotopic lapse rate” will not exist.

In addition, with regard to making the difference clear the definitions of the IAE in water vapor and

the conventional IAE in precipitation (or other surface isotopic carriers): Following your suggestions, we have added several sentences to clarify the difference between them, i.e., “*We note in this study, the definition of the IAE in water vapor is slightly different from the conventional IAE in precipitation or other surface isotopic carriers that traditionally focus on the relationship between isotopes and topography along a mountain range (different locations with increasing altitude). Here we define the IAE in water vapor to describe the relationships between δD_v and different atmospheric pressure levels from the same location (same location with increasing altitude), i.e., the δD_v increases with increasing altitude in the atmosphere from the lower level to the upper level.*” Please see L388-L395 in the revised manuscript.

2) With regard to the connection between the isotope-topography relationship and the relationship in the free troposphere: Following your suggestions, we added a new figure (Fig. 4 in the revised manuscript) and a new section “**Connection between the IAE in water vapor and precipitation**” to clarify the connection between the isotope-topography relationship and the relationship in the free troposphere, i.e., “**Connection between the IAE in water vapor and precipitation.** *As water vapor acts as the “mass source” of precipitation, the stable isotopic composition of water vapor will directly influence the stable isotopic composition of precipitation. Our results indicated that, in the mountainous regions, the patterns of the stable isotopic composition of water vapor at different atmospheric pressure levels govern those of the corresponding precipitation, via advection (Fig. 4). Hence, the IAE in water vapor will be imprinted on precipitation (Fig. 4). Taking the WUSA as an example, at the 618 hPa level, air masses laterally mix with moisture containing higher isotope values along the moisture transport pathway and this signal is preserved through advection processes which results in higher isotope values in precipitation at the target 618 hPa level. In contrast, at the 681 hPa level, the relatively higher isotopic signal in water vapor becomes relatively depleted along the moisture transport pathway due to weak vertical mixing with the lower-troposphere (Supplementary Table 2). This process results in relatively depleted isotope values in precipitation at the 681 hPa level compared to the 618 hPa level, although isotope values in precipitation at the 681 hPa level are still higher than at the lower levels (Fig. 4A). As a consequence, the isotope values in water vapor increase with increasing altitude in the atmosphere from the lower level to the upper level, which produces the IAE in water vapor. Similarly, the isotope values in corresponding precipitation will increase with increasing altitude from the lower topography to the higher topography, and the IAE occurs in corresponding precipitation. Hence, precipitation inherits the IAE in water vapor. Similar processes can be used to explain the IAE in water vapor and precipitation in the NTP (Fig. 4B). The spatial distributions of the IAE reported in precipitation¹⁶⁻¹⁸ and other surface isotopic carriers¹⁹⁻²⁶ on the global scale (Fig. 1) are mostly consistent with the occurrence of the IAE in water vapor which further demonstrates the close coupling between the stable isotope signals of water vapor and precipitation. It is evident that the IAE in water vapor determines the IAE of precipitation before the influence of localized factors may take part.*” Please see L306-L333 in the revised manuscript.

Fig. 4. Schematic diagrams of the mechanisms that cause the IAE from an atmospheric circulation perspective. (A) WUSA, western United States of America. (B) NTP, northern Tibetan Plateau. Seas include the Mediterranean Sea, Red Sea, Persian Gulf, and Caspian Sea. The gray dots in air masses represent the intensity of the lateral mixing, i.e., denser dots, stronger the lateral mixing.

3) With regard to whether the IAE exists at the sub-cloud level: We rechecked whether the IAE exists at the sub-cloud level. The results showed that the IAE is very weak under the 825 hPa level. Hence, we mainly focused on the IAE above 825 hPa in the main text. Following your comments, we updated Supplementary Fig. 1 and Supplementary Fig. 2 to show the results under the 825 hPa level. Please see the new Supplementary Fig. 1 and Supplementary Fig. 2 in the revised Supplementary Information.

Supplementary Fig. 1. Seasonal spatial patterns of the IAE at different atmospheric levels at the lower and mid-latitudes. (A-E) spring (March-April-May: MAM). (F-J) summer (June-July-August: JJA). The black boxes show the WUSA region (31° N – 45° N, 118° W – 106° W) while the blue boxes indicate the NTP region (35° N – 45° N, 66° E – 106° E). The IAE is very weak across the globe under the 825 hPa level. Hence, in this study we mainly discuss the IAE above the 825 hPa level.

Supplementary Fig. 2. Seasonal spatial patterns of the IAE at different atmospheric levels at the lower and mid-latitudes. (A-E) autumn (September-October-November: SON). (F-J) winter (December-January-February: DJF). The black boxes show the WUSA region (31° N – 45° N, 118° W – 106° W) while the blue boxes indicate the NTP region (35° N – 45° N, 66° E – 106° E). The IAE is very weak under the 825 hPa level across the globe. Hence, in this study we mainly discuss the IAE above the 825 hPa level.

My second concern is the correct use of “stable isotopes” in the paper. The introduction part is largely about previous observations and modelling studies on $\delta^{18}\text{O}$ in precipitation ($\delta^{18}\text{Op}$). However, the study of Jing et al., is about δD in vapour. I would suggest the authors indicate the link between those isotopic species that is not evident for broad audience of the Nature Communications. I suggest that the authors specify exactly which isotopes are involved in the study already the abstract.

Response: Thank you for pointing this issue out. As variations of δD are approximately 8 times larger than $\delta^{18}O$, we can apply δD_v to identify the key processes responsible for the IAE in precipitation $\delta^{18}O$ or other carriers that are rooted in precipitation. Following your suggestion, we specified it as hydrogen isotope (δD , that was used in this study) throughout this manuscript. In the abstract, we changed “The “inverse altitude effect” (IAE) where stable isotopes increase with increasing altitude, directly contradicts the basic theory of paleoaltimetry which require stable isotopic ratios to be primarily influenced by the altitude effect. We explore the cause(s) of the IAE in water vapor...” to “The IAE directly contradicts the basic theory of stable isotope paleoaltimetry. However, the causes of the IAE remain unclear. Here, we explore the mechanisms of the IAE from an atmospheric circulation perspective using δD in water vapor on a global scale.” Please see L28-L31 in the revised manuscript.

Moreover, we provide a detailed explanation of the link between $\delta^{18}O$ and δD in our manuscript, i.e., “Under different environmental conditions at a specific site, δD_v and $\delta^{18}O_v$ generally follow the curves of isotopes in meteoric water³¹⁻³⁴. The variations of δD are approximately 8 times larger than $\delta^{18}O$. Therefore, we can apply satellite-derived δD_v to identify the key processes responsible for the IAE identified in precipitation $\delta^{18}O$ and other surface isotopic carriers that are rooted in precipitation.” Please see L92-L96 in the revised manuscript.

References:

31. Yu, W., Tian, L., Ma, Y., Xu, B. & Qu, D. Simultaneous monitoring of stable oxygen isotope composition in water vapour and precipitation over the central Tibetan Plateau. *Atmos. Chem. Phys.* **15**, 10251-10262 (2015).
32. Bony, S., Risi, C., & Vimeux, F. Influence of convective processes on the isotopic composition ($\delta^{18}O$ and δD) of precipitation and water vapor in the tropics: 1. Radiative-convective equilibrium and Tropical Ocean-Global Atmosphere-Coupled Ocean-Atmosphere Response Experiment (TOGA-COARE) simulations. *J. Geophys. Res.-Atmos.* **113**, D19305 (2008).
33. Li, Y. et al. Variations of stable isotopic composition in atmospheric water vapor and their controlling factors—A 6-year continuous sampling study in Nanjing, eastern China. *J. Geophys. Res.-Atmos.* **125**, e2019JD031697 (2020).
34. Bonne, J.-L. et al. The isotopic composition of water vapour and precipitation in Ivittuut, southern Greenland. *Atmos. Chem. Phys.* **14**, 4419-4439 (2014).

Third, I am confused by the term “moisture source”. It is common to trace the “moisture source regions” or “moisture transport pathways” to understand the isotopic signature in vapour (e. g., Sodemann et al., 2009; Dahinden et al., 2021). However, as far as I understand, the authors do not use the term “moisture source” for a geographic region of water uptake, but for different vertical levels of the troposphere (e.g., lines 122-125; Table 1). Isn't this point about moisture uptake at different levels more relevant to prove the increase in vertical mixing in the regions of interest? What about the lateral geographic regions of moisture transport? Maybe it's just me, but I was not clear on what the term “moisture source” actually meant in this study, please explain in more detail in the main text. I find

the author's point regarding the change in moisture sources crucial to understanding the paper, as it provides a mechanism to explain the IAE. I would suggest showing the results of the HYPSPPLIT trajectory analysis in the main text?

Response: Thank you for pointing these issues out. We addressed them as follows.

1) With regard to the term "moisture source": We are sorry that this expression caused confusion. In this study, "moisture source" is used as a relatively broad concept, which includes both the "moisture source regions" and "moisture transport pathways."

First, with regard to the term "moisture source regions": In this study, we used atmospheric circulation patterns to analyze the moisture source regions and reveal the influence of the moisture from such "source regions" on the IAE. Please see Fig. 4 in the revised manuscript. Taking the WUSA as an example, it is clear that the oceanic moisture with higher isotopic values at 600 hPa level over the tropical Atlantic is transported into the WUSA during summer (Fig. 4). Combining the atmospheric circulation patterns with δD_v patterns, we can conclude that the tropical Atlantic is the "moisture source region" where the moisture contributes to the IAE over the WUSA. For the NTP, the "moisture source regions" where the moisture contributes to the IAE are the regions of the Mediterranean Sea, Red Sea, Persian Gulf, and Caspian Sea.

Second, with regard to the term "moisture transport pathways": In this study, we discussed the impact of "moisture transport pathways" on the IAE over the target regions (the WUSA and the NTP) by analyzing the moisture contribution from different atmospheric levels over the moisture source regions to the different levels over the target regions. Taking the WUSA as an example, the moisture transported from the lower troposphere (below 825 hPa) contributes 89% moisture to the target 825 hPa and 73% to target 750 hPa levels (Supplementary Table 2), which demonstrates that the main moisture transport pathway for the target 825 hPa and 750 hPa levels is characterized by an "upslope" type (Fig. 4A). In contrast, the moisture transport pathways are relatively complex from the moisture source regions to the target 681 and 618 hPa levels over the WUSA. While the moisture enroute in the lower troposphere contributes 49% and 32% moisture to the target 681 hPa and 618 hPa levels, respectively, the moisture contribution enroute from the mid-troposphere becomes the largest (Supplementary Table 2), which indicates that the dominant moisture transport pathway for the target 681 hPa and 618 hPa levels is characterized by an "advection" type (Fig. 4A).

Following your comments, we added one sentence to clarify that the term "moisture source" is meant to include the "moisture source regions" and the "moisture transport pathways", i.e., *Note in this study, we use the term moisture source as a relatively broad concept, which includes both the moisture source region and the moisture transport pathway.* Please see L447-L449 in the revised manuscript.

In addition, we have clarified the term throughout the manuscript. Please see L34, L100, L154, L159, L163, L164, L169, L237, L239, L243, L246, L248, L257, L297, L314, L317, L337, L344 and L360 in the revised manuscript

More importantly, we have rephrased the statements of "moisture transport pathways" for both the WUSA and the NTP. Please see L155- L160, L168-L171, and L237-L240 in the revised manuscript

Reference:

Lee, J.-E., Risi, C., Fung, I., Worden, J., Scheepmaker, R. A., Lintner, B., Frankenberg, C. Asian monsoon hydrometeorology from TES and SCIAMACHY water vapor isotope measurements and LMDZ simulations: Implications for speleothem climate record interpretation. *J. Geophys. Res.*, 2012, 117, D15112, doi:10.1029/2011JD017133.

2) With regard to the HYPSPPLIT trajectory analysis: Following your suggestion, we added two new figures to show the results of the HYSPLIT trajectory analysis (Supplementary Fig. 7 and Supplementary Fig. 13). Taking the WUSA as example, the results of the HYSPLIT trajectory analysis show that a moisture channel derived from the Atlantic anticyclone emerges during summer (Supplementary Fig. 7 F-J). However, the moisture channel is very weak during the other three seasons. The results of the HYSPLIT trajectory analysis are similar to the atmospheric circulation pattern analysis (Fig. 4). Compared with the trajectory analysis, the atmospheric circulation analysis provides a better view of an obvious clockwise transport channel from the tropical Atlantic to Mexico and then to the WUSA (Fig. 4). Moreover, in our manuscript, we have already provided a detailed discussion about the role of the tropical Atlantic Ocean on the IAE by analyzing the atmospheric circulation fields. To avoid repetition, we have only added one additional sentence in the main text to describe the results of the trajectory analysis for the WUSA, i.e., “*The trajectory frequency analyses also confirm the existence of a tropical Atlantic-originated moisture channel that is operational during summer (Supplementary Fig. 7).*” and added the outputs of the trajectory analysis (the Supplementary Fig. 7) into the revised Supplementary Information.

Please see L196-L198 and L262-L263 in the revised manuscript and the new Supplementary Fig. 7 and Supplementary Fig. 13 in the revised Supplementary Information.

Supplementary Fig. 7. Seasonal trajectory frequencies for different levels in the WUSA during 2006-2009. (A-E) spring (MAM). (F-J) summer (JJA). (K-O) autumn (SON). (P-T) winter (DJF).

3) With regard to “Isn’t this point about moisture uptake at different levels more relevant to prove the increase in vertical mixing in the regions of interest? What about the lateral geographic regions of moisture transport?”: Thank you for pointing these issues out. We addressed them as follows.

a) In Table 1, we do not refer to the local vertical mixing, but rather the lateral mixing between the lower-troposphere and the mid-troposphere along moisture transport pathway. Please refer to the schematic diagram (Fig. 4 in the revised manuscript). Following your comments, we clarify whether the mixing is “the lateral mixing” or “the local vertical mixing.” Please see L33, L170, L179, L184, L224, L242, L250, L255, L318, L337, L339, L343, L354, and L698 in the revised manuscript.

b) Of course, we cannot completely exclude the existence of local vertical mixing. However, it can be seen from the results of this study and the schematic diagram that the local vertical mixing can be negligible. If vertical mixing of water vapor occurs, we will find that the isotopes in water vapor will be lower when local water vapor in the lower troposphere is transported upward to the mid-troposphere where the temperature is lower. Clearly, the consequence will weaken the IAE and even lead to the disappearance of the IAE. Similarly, when upper-level water vapor without high isotope values locally mixes downwards, the consequence will also weaken the IAE or even cause it to disappear. The reason the upper-level water vapor can maintain high isotope values is precisely dependent on the lateral mixing (Fig. 4) along the moisture transport pathway. Indeed, the occurrence of the IAE in our study proves that local vertical mixing is insignificant. Please refer to the schematic diagram (Fig. 4). Following your comments, we clarified that the contribution of the “local vertical mixing” can be negligible. Please see L170 and L318 in the revised manuscript.

c) About the lateral geographic regions of moisture transport: It seems that this issue is similar to the issue about the “moisture source regions”. As mentioned above, in this study, we used atmospheric circulation patterns to analyze the lateral geographic regions of moisture transport and reveal the influence of the moisture from lateral geographic regions on the IAE. Please see Fig. 4 in the manuscript.

Finally, the authors attribute the observed IAE to two processes: (1) the supply of moisture with higher isotopic values from source regions, and (2) intense mixing between the lower and mid-troposphere during moisture transport. However, the impact of other processes that might also have a contribution has not been evaluated, only mentioned (lines 65-67). How much these effects, and others (?) contribute to the detected relationships? Please see Risi et al., 2019 (Atmos. Chem. Phys.) for a decomposition method.

Response: Thank you for pointing this issue out. The decomposition method to examine potential local factors is likely not suitable for this study for the following reasons.

1) Such local factors, like the re-evaporation of raindrops, the exchange of raindrops with surrounding water vapor and post-condensation processes all can affect the IAE of the surface isotopic carriers from a local perspective. However, these local factors are diverse and their effects are difficult to reconcile. Clearly, it is more complicated to consider the influences of such local factors on the IAE. Hence, we chose a unique isotope carrier above the surface, i.e., water vapor, to discuss the cause(s) of the IAE, rather than the surface isotope carriers. Unlike the surface isotopic carriers, the IAE in water vapor is less affected by local factors such as sub-cloud evaporation, post-depositional processes, or local moisture recycling that may have influenced other IAE measurements reported

from precipitation and other isotopic records. Moreover, our results show that the IAE mainly occurs at the 618 hPa level, where isotopes are less influenced by such local factors.

2) As mentioned above, those local factors are diverse and their influences on the IAE are difficult to reconcile. In this study, we try to seek the key factors that govern the IAE in water vapor on a broader spatial scale, rather than investigate local factors. Our findings show that, in both regions of the WUSA and the NTP, the supply of moisture with higher isotopic values from distant source regions, and intense lateral mixing between the lower and mid-troposphere along moisture transport pathway contribute to the IAE in water vapor. This conclusion is universal and may also be used to explain the IAE in water vapor for other regions, such as the Andes, South Africa and Australia. If we discuss the influence of local factors again, we are afraid that the innovation of this paper will be greatly weakened.

Following your suggestion, we added two sentences to explain why we focused on the key factors that govern the IAE in water vapor on broader spatial scales, rather than on the local factors, i.e., “*These issues highlight the need to better understand the causes of the IAE, especially over larger spatial and temporal scales*” and “*In particular, the δD_v and $\delta^{18}O_v$ above the sub-cloud level is less affected by localized factors that may influence other IAE measurements reported from precipitation and snow records. As such, satellite measurements can be used to independently analyze the influence of large-scale atmospheric circulations on the IAE.*” Please see L83-L84 and L87-L91 in the revised manuscript.

Minor points

Line 37. The authors might prefer to refer the latest work on the Alpine paleoaltimetry by Krsnik et al. 2021 (Solid Earth), which builds on the older study by Campany et al. 2012.

Response: Following your suggestion, we refer to the latest work on the Alpine paleoaltimetry by Krsnik et al. 2021 (Solid Earth). Please see L41 in the revised manuscript.

Line 57. Some additional references missing from the Introduction: Levin et al., 2009, J. Geophys. Res. Atmos.) and Rohrmann et al. 2014 (EPSL). Both papers found flat isotopic lapse rates.

Response: Following your suggestion, we added these two references, i.e., Ref. 16 and Ref. 21. Please see L60 in the revised manuscript.

Line 67. In the study by Levin et al. 2009, the unusual isotopic-altitude relationship was attributed to convective instability in areas of high convective precipitation.

Response: Following your suggestion, we added another factor (convective instability), and added the reference of Levin et al., 2009. Please see L76 in the revised manuscript.

Lines 74-75. Please refer to global studies or studies for different geographic regions, not just for the Tibet

Response: Following your suggestion, we added several references, which study regions covers the tropics, the monsoon region and the polar region. Please see L93 in the revised manuscript.

Line 75. Delete “moreover”.

Response: We deleted “moreover”. Please see L93 in the revised manuscript.

Line 79. Why excluding sub-cloud level?

Response: Following your comments, we have now included the sub-cloud level in this revised manuscript. We updated the Supplementary Fig. 1 and Supplementary Fig.2 to show the results under the 825 hPa level. Please see the new Supplementary Fig. 1 and Supplementary Fig.2 in the revised Supplementary Information.

Line 83. I was confused several times while reading, is it vertical transport or advection? Please specify here and elsewhere in the text.

Response: In this study, moisture transport includes both vertical moisture transport (such as upslope) and advection. Similarly, the mixing includes both the local vertical mixing and the lateral mixing between the lower and mid-troposphere along moisture transport pathway. Please see the schematic diagram of the mechanisms of the IAE from an atmospheric circulation perspective (Figure 4). Following your suggestion, we have specified them here and throughout the manuscript. Please see L33, L170, L179, L184, L224, L242, L250, L255, L318, L337, L339, L343, L354, and L698 in the revised manuscript.

Lines 125-126. Does this rapid vertical exchange occur everywhere on the continents? Please reference global studies, not a study for China.

Response: Frankly speaking, we are not sure whether this rapid vertical exchange occurs everywhere on the continents, except for the study area in China. Taking the WUSA as an example, our results show that the lower troposphere (below 825 hPa) contributes 89% moisture to 825 hPa and 73% to 750 hPa (Table 1). It implies that the moisture at 825 hPa and 750 hPa is dominated by vertical moisture transport from the lower altitudes. Following your comments, we changed “Our results show that the lower troposphere (below 825 hPa) contributes 89% moisture to 825 hPa and 73% to 750 hPa (Table 1). The lower troposphere is dominated by rapid vertical exchanges of moisture between the Earth’s surface and the free troposphere³⁵” to “*Our results show that the moisture transported from the lower troposphere (below 825 hPa) contributes 89% moisture to the target 825 hPa level and 73% to the target 750 hPa level (Supplementary Table 2), which demonstrates that the main moisture transport pathway for the target 825 and 750 hPa levels is characterized by an “upslope” type (Fig. 4A).*” Please see L155-L160 in the revised manuscript.

Lines 129-130. Replace “simulated” with “described” or similar.

Response: We replaced “simulated” with “described”. Please see L162 in the revised manuscript.

Lines 264-265. Does isotopic composition in the free troposphere play a role in precipitation composition? Can you show this by correlation of so?

Response: Yes, isotopic composition in the free troposphere plays a role in precipitation composition. Following your comments, we added additional sentences to the new section “Connection between the IAE in water vapor and precipitation” to discuss the influence of isotopic composition in the free troposphere on precipitation composition. Please see L306-333 in the revised manuscript.

Frankly speaking, in this study we only discuss the qualitative relationship between isotopic composition in the free troposphere and precipitation composition and cannot show a quantitative relationship between them.

Lines 274-275. Better “Uncertainties in the parameterization”?

Response: Following your suggestion, we changed “uncertainties related to” to “uncertainties in the parameterization of”. Please see L365 in the revised manuscript.

Lines 276-277. What is about the mixing between the boundary layer and the free troposphere?

Response: We are sorry for the confusion. Here we are discussing the air mixing between the lower and mid-troposphere. Following your comments, we changed the “*the boundary layer and free atmosphere*” to “*the lower and mid-troposphere*” to keep consistent with the expression throughout the manuscript. Please see L366 in the revised manuscript.

Sincerely,
Svetlana Botsyun

REVIEWERS' COMMENTS

Reviewer #1 (Remarks to the Author):

After reading the revised manuscript, I can find the authors had given reasonable responses to the reviewers' comments. The revision was well organized. New figures and paragraphs were added to explain the main findings of the manuscript. To make the innovation of the article more readable, I suggest that:

- 1) For "Fig.4. Schematic diagrams of the mechanisms that cause the IAE from an atmospheric circulation perspective". The current figure has explained the mechanisms of water vapor mixing process clearly but lack of data to show the IAE and the contribution of different moisture sources. Therefore, firstly, please add the average isotopic value of each level of water vapor at specific atmospheric pressure to show the IAE phenomenon directly by data. Secondly, please add the contributing percentage of each moisture transporting branches from the lower or upper troposphere to the target levels, since the authors already calculated the relative data. For example, "the WUSA, the moisture transported from the lower troposphere (below 825 hPa) contributes 89% moisture to the target 825 hPa and 73% to target 750 hPa levels."
- 2) The manuscript had analyzed the seasonal isotopic variation of atmosphere vapor of spring, summer, autumn and winter, but the current version used a self-made acronym such as "spring (MAM), summer (JJA), autumn (SON), winter (DJF)" to describe them. I suggest to use the season's noun itself to reduce confusion.
- 3) Line 322, 324, There are two "increase" in the sentence, its' confusing, delete one of it or revise it.
- 4) Line 332-333 Perhaps sentence "It is evident that the IAE in water vapor determines the IAE of precipitation before the influence of localized factors may take part" needs to be clarify further, because the vertical mixing process has the potential to transport the local evaporation vapor to the higher level (this can be seen from Fig. 4) . At the same time, the spatial and temporal heterogeneity of AE and IAE are impacted by the mixtures of multiple moistures sources, especially the dominant sources (see Jiao et al,2020, Journal of Hydrology).

Reviewer #2 (Remarks to the Author):

Dear authors,

thank you very much for revising your manuscript and for all your detailed answers to my comments. The manuscript became more clear to me, is now well structured and it has gained in significance. Thank you especially for including the new figure 4, that helps the reader to easier understand the topic. It is really important to have a deeper knowledge on the IAE and I encourage you to continue with your research.

I recommend this manuscript for publication without further revisions.

Sincerely,

Dr. Christiane Meier

Reviewer #3 (Remarks to the Author):

I would like to highlight again high value of this manuscript to the paleoaltimetry community. I found the responses to my comments reasonable and thorough and the revised version of the manuscript substantially improved. I have no further points to add and recommend the article for prompt publication.

Sincerely,

Svetlana Botsyun

Replies to Reviewers

(Reviewers' statements in black, and our response in blue)

Line 224, -the line 224 in the previous manuscript.

L83-L84, -the lines 83-84 in the revision.

REVIEWERS' COMMENTS

Reviewer #1 (Remarks to the Author):

After reading the revised manuscript, I can find the authors had given reasonable responses to the reviewers' comments. The revision was well organized. New figures and paragraphs were added to explain the main findings of the manuscript. To make the innovation of the article more readable, I suggest that:

1) For “Fig.4. Schematic diagrams of the mechanisms that cause the IAE from an atmospheric circulation perspective” . The current figure has explained the mechanisms of water vapor mixing process clearly but lack of data to show the IAE and the contribution of different moisture sources. Therefore, firstly, please add the average isotopic value of each level of water vapor at specific atmospheric pressure to show the IAE phenomenon directly by data. Secondly, please add the contributing percentage of each moisture transporting branches from the lower or upper troposphere to the target levels, since the authors already calculated the relative data. For example, “the WUSA, the moisture transported from the lower troposphere (below 825 hPa) contributes 89% moisture to the target 825 hPa and 73% to target 750 hPa levels.”

Response: Thank you for this suggestion. We did try to include the average water vapor isotopic values for each atmospheric pressure level and also add the corresponding contribution percentages. However, we found that the presentation of these alternative schematic diagrams became much too busy and complicated. In fact, we had already included several symbols to represent the IAE and the relative contributions within the existing schematic diagrams. For example, the plus (minus) signs within the circles indicate the inverse altitude effect (IAE) (altitude effect: AE), and the sizes of the circles represent the significance of the IAE (AE), i.e., larger circles represent a clearer IAE (AE) presence. In addition, we used the white arrows to indicate the moisture contribution from the mid-troposphere (above 825 hPa) to the target region and the gray arrows to indicate the moisture contribution from the lower troposphere (below 825 hPa) to the target region. The different sizes of the arrows represent the relative moisture contribution percentages. In each panel, red dashed ellipses mark the levels (or altitudes) where the IAE occurs.

To keep the schematic diagrams more concise, we did not add such values on the figure. However, we now include additional sentences to the figure caption that better explain the symbols to address Reviewer #1's concerns, i.e., “*Note the plus (minus) signs within the circles indicate the inverse altitude effect (altitude effect), and the sizes of the circles represent the strength of the IAE (AE), i.e., larger circles represent a more pronounced IAE (AE). The white arrows indicate the moisture contributions from the mid-troposphere (above 825 hPa) to the target region. The gray arrows*

indicate the moisture contributions from the lower troposphere (below 825 hPa) to the target region. The different sizes of the arrows represent the relative moisture contribution percentages. In each panel, red dashed ellipses mark the levels (or altitudes) where the IAE occurs.” Please see L677-685 in the revised manuscript.

2) The manuscript had analyzed the seasonal isotopic variation of atmosphere vapor of spring, summer, autumn and winter, but the current version used a self-made acronym such as “spring (MAM), summer (JJA), autumn (SON), winter (DJF)” to describe them. I suggest to use the season’s noun itself to reduce confusion.

Response: In general, when we talk about the summer, it refers to the JJA for the Northern Hemisphere while it refers to DJF for the Southern Hemisphere. Hence, to avoid this ambiguity, we used the MAM, JJA, SON, and DJF to refer to the spring, summer, autumn and winter, respectively. Moreover, those acronyms are widely used in research articles.

Following your suggestion, we provide the full names of those acronyms the first time they appear in the main text. Moreover, we provide the full names of those acronyms in each figure to avoid potential confusion. Please see L113, L125, and L127 in the revised manuscript.

3) Line 322, 324, There are two “increase” in the sentence, its’ confusing, delete one of it or revise it.

Response: Thank you for pointing this issue out. We changed “will increase with increasing altitude” to “will increase with altitude”. Please see L323 and L325 in the revised manuscript.

4) Line 332-333 Perhaps sentence “It is evident that the IAE in water vapor determines the IAE of precipitation before the influence of localized factors may take part” needs to be clarify further, because the vertical mixing process has the potential to transport the local evaporation vapor to the higher level (this can be seen from Fig. 4). At the same time, the spatial and temporal heterogeneity of AE and IAE are impacted by the mixtures of multiple moistures sources, especially the dominant sources (see Jiao et al,2020, Journal of Hydrology).

Response: Thank you for pointing these two issues out. We have addressed them as follows.

1) With regard to “the vertical mixing process has the potential to transport the local evaporation vapor to the higher level”, we found that your comment is similar to that raised by Reviewer #3 in the previous review. We would like to explain once more as follows.

If vertical mixing of water vapor occurs, we will find that the isotopes in water vapor will be lower when local water vapor in the lower troposphere is transported upward to the mid-troposphere where the temperature is lower. Hence, such mixing will weaken the IAE and even lead to its disappearance. Other localized factors, like the re-evaporation of raindrops, the exchange of raindrops with the surrounding water vapor and post-condensation processes all can affect the IAE of the surface isotopic carriers from a local perspective. However, these local factors are diverse and their effects are difficult to reconcile. Clearly, it is considerably more complicated to understand and quantify the influences of such local factors on the IAE.

Hence, we chose a unique isotope carrier above the surface, i.e., water vapor, to discuss the cause(s) of the IAE, rather than the surface isotope carriers, like precipitation. Unlike the surface isotopic carriers, the IAE in water vapor is less affected by local factors such as sub-cloud evaporation, post-depositional processes, or local moisture recycling that may have influenced other IAE measurements reported from precipitation and other isotopic records. Moreover, our results show that the IAE mainly occurs at the 618 hPa level, where isotopes are less influenced by such local factors. More importantly, in this manuscript, we mainly focus on the causes of the IAE over broader spatial and temporal scales and emphasized that the supply of moisture with higher isotopic values from distant source regions, and intense lateral mixing between the lower and mid-troposphere along the moisture transport pathway contribute to the IAE in water vapor. This conclusion is universal and may also be used to explain the IAE in water vapor for other regions, such as the Andes, South Africa and Australia. If we discuss the influence of local factors again, we are afraid that the innovation of this paper will be greatly weakened.

To address this issue, during the last review and revision, we included additional sentences to explain why we focused on the key factors that govern the IAE in water vapor on broader spatial scales, rather than on the local factors, i.e., *“These issues highlight the need to better understand the causes of the IAE, especially over larger spatial and temporal scales”* and *“In particular, the δD_v and $\delta^{18}O_v$ above the sub-cloud level are less affected by localized factors that may influence other IAE measurements reported from precipitation and snow records. As such, satellite measurements can be used to independently analyze the influence of large-scale atmospheric circulations on the IAE.”* Please see L83-L84 and L87-L91 in the revised manuscript.

2) With regard to “the spatial and temporal heterogeneity of IAE are impacted by the mixtures of multiple moistures sources, especially the dominant sources (see Jiao et al, 2020, Journal of Hydrology)”, Yes, you are right. As mentioned in Jiao et al. (2020), the influence of the mixtures of multiple moisture sources is very important, and the mixtures of multiple moisture sources can lead to no elevation effect (i.e. the occurrence of the IAE). Indeed, the finding of Jiao et al. (2020) who used local data is consistent with our regional/global data presented in this study. Following your comment, we have added the Jian et al. (2020) reference to our manuscript. Please see L60 in the revised manuscript.

Considering the both

Reviewer #2 (Remarks to the Author):

Dear authors,

thank you very much for revising your manuscript and for all your detailed answers to my comments. The manuscript became more clear to me, is now well structured and it has gained in significance. Thank you especially for including the new figure 4, that helps the reader to easier understand the

topic. It is really important to have a deeper knowledge on the IAE and I encourage you to continue with your research.

I recommend this manuscript for publication without further revisions.

Sincerely,

Dr. Christiane Meier

Response: Thank you so much for your constructive comments and suggestions, which have considerably improved our manuscript.

Reviewer #3 (Remarks to the Author):

I would like to highlight again high value of this manuscript to the paleoaltimetry community. I found the responses to my comments reasonable and thorough and the revised version of the manuscript substantially improved. I have no further points to add and recommend the article for prompt publication.

Sincerely,

Svetlana Botsyun

Response: Thank you so much for your constructive comments and suggestions, which have considerably improved our manuscript.